# Scaled Gradient Mean Subtraction: A Lightweight Method for Amplifying Underutilized Gradient Directions

## Abstract

We propose Scaled Gradient Mean Subtraction (SGMS), a lightweight method that improves neural network training by amplifying underutilized gradient directions. In mini-batch training, gradients from individual samples are expected to point in diverse directions, but in practice they are often highly correlated, spanning only a few dominant directions. Consequently, weight updates are confined to a low-rank subspace, leaving many directions underutilized. SGMS addresses this imbalance by subtracting a scaled mean from the gradient. With common ReLU-like activations (e.g., ReLU, GeLU, SiLU), this simple operation weakens the most dominant direction, allowing complementary directions to play a greater role in optimization. Unlike approaches that rely on costly covariance statistics or matrix decompositions, SGMS achieves a similar rebalancing of gradient directions with a single mean-subtraction step, adding minimal overhead and requiring no architectural changes. Formally, SGMS generalizes Gradient Centralization (GC) as a special case, but—by partially rather than fully suppressing the mean direction—it retains valuable gradient components that GC eliminates. Experiments on CIFAR and ImageNet show that SGMS brings substantial improvements on Convolutional Networks (ResNets), and small but consistent gains on Transformers and a GPT-style language model, while adding negligible overhead.

## 1 Introduction

Deep neural networks (DNNs) have achieved remarkable success across a wide range of tasks, including image recognition, natural language processing, and speech recognition. However, these advances are largely driven by increasing model scale, which makes both training and deployment computationally demanding. This has motivated research on improving training efficiency.

One of the fundamental reasons why DNN training can be inefficient lies in the structure of gradient updates. In mini-batch stochastic gradient descent, gradients from different samples are ideally expected to point in diverse directions. In practice, however, they are often strongly correlated and span only a few dominant directions in parameter space. This phenomenon has also been noted in prior work (Hu et al., 2022; Zhao et al., 2024), which exploits low-rank structures in gradients. This *low-rank subspace concentration* leaves many potentially useful directions underutilized. *Amplifying these underutilized gradient directions* is therefore a natural step toward more efficient training.

We propose *Scaled Gradient Mean Subtraction (SGMS)*, a lightweight method that improves training performance by amplifying underutilized gradient directions. SGMS simply subtracts a scaled mean from the gradients. When combined with common nonnegative activations (e.g., ReLU), this operation directly weakens the most dominant direction of gradients, allowing complementary directions to play a greater role in optimization. Typically, achieving such balancing of gradient directions requires costly matrix decompositions such as singular value decomposition (SVD). Notably, SGMS attains a similar effect through a single mean subtraction, adding minimal (often negligible) overhead and requiring no architectural changes. Moreover, SGMS is orthogonal to optimizer design: it can be used as a plug-in for optimizers such as SGD and Adam, and is neither a competitor nor a replacement for them.

Importantly, SGMS provides a broader perspective on *Gradient Centralization (GC)* (Yong et al., 2020). Fully subtracting the mean corresponds exactly to GC, which can be interpreted as completely suppressing the dominant gradient direction. While this may accelerate convergence, it also risks discarding useful signal in the suppressed direction. By contrast, SGMS allows partial suppression, preserving valuable gradient direction while still amplifying weaker directions, thereby offering a more flexible middle ground.

Related approaches pursue the same goal of balancing gradient directions using preconditioning techniques that decorrelate gradients through matrix decompositions. AdaBK (Yong et al., 2023) is a representative example: it computes layer-wise second-moment estimates and constructs approximate inverse-root preconditioners, effectively decorrelating gradient updates and expanding the range of directions that contribute to learning. While effective against low-rank subspace concentration, such methods incur nontrivial computation and memory overhead in training. SGMS offers a lightweight alternative, achieving a similar rebalancing effect with only a single mean subtraction and minimal (often negligible) overhead.

We evaluate SGMS on CIFAR-10/100 and ImageNet-1K across multiple architectures, observing substantial improvements on CNNs and small but consistent gains on Transformers, all with minimal overhead. Our contributions are:

- We revisit subspace concentration of gradient updates as a key inefficiency factor and motivate the goal of amplifying underutilized gradient directions.
- We propose SGMS, a lightweight method that weakens dominant gradient directions via a simple mean-subtraction instead of costly matrix decompositions or iterative methods.
- We clarify the connection to Gradient Centralization (GC)-SGMS reduces to GC at its endpoint-and show that SGMS occupies a practical position between GC and covariance-based preconditioners such as AdaBK in terms of cost–benefit trade-offs.
- We demonstrate on CIFAR and ImageNet that SGMS improves accuracy while adding minimal overhead.

## 2 RELATED WORK

We review three areas of research most relevant to SGMS: (i) optimization methods for DNN training, (ii) approaches leveraging subspace structures in parameter space, and (iii) techniques that mitigate subspace concentration by modifying gradient directions.

**Optimization methods.** The foundational method for training DNNs is stochastic gradient descent (SGD) (Robbins & Monro, 1951), with popular variants such as momentum (Sutskever et al., 2013) and Nesterov acceleration (Nesterov, 1983). Adaptive methods, including RMSProp (Tieleman & Hinton, 2012), AdaDelta (Zeiler, 2012) and Adam (Kingma & Ba, 2015), adjust update magnitudes based on statistics of past gradients, and many extensions have been proposed (Loshchilov & Hutter, 2019; Dozat, 2016; Liu et al., 2020; Zhuang et al., 2020). These optimizers improve training efficiency but operate on a per-parameter basis, ignoring correlations among gradient directions, and therefore do not mitigate the subspace concentration problem. SGMS is orthogonal to optimizer design and can be combined with any standard optimizer as a plug-in component to alleviate subspace concentration.

**Approaches leveraging subspace structures.** Several methods exploit the low-rank structure of gradients for parameter-efficient training, most notably LoRA (Hu et al., 2022) and its variants (Dettmers et al., 2023; Zhang et al., 2023; Valipour et al., 2023). Such approaches restrict updates to a low-dimensional subspace spanned by dominant directions. In contrast, SGMS takes the opposite perspective: rather than confining updates to the dominant subspace, it amplifies complementary directions that remain underutilized in standard training, thereby aiming to improve generalization at minimal overhead.

**Gradient centralization and preconditioners.** Gradient Centralization (GC) (Yong et al., 2020) can be viewed as the special case of SGMS that fully subtracts the mean from gradients, projecting them onto the zero-mean subspace. Geometrically, with nonnegative activations the mean gradient

direction typically aligns with the dominant update direction, so GC effectively suppresses this direction entirely. While this may accelerate convergence, it also risks discarding useful information in that direction. Note that the beneficial effects of gradient centering have also been discussed in other prior work (LeCun et al., 1991; Schraudolph, 1998a; LeCun, 2015), and the element-wise centering effect is also discussed in (Wiesler et al., 2014). While related work has discussed centralization (Schraudolph, 1998b; Montavon & Müller, 2012; Yong et al., 2020), none links mean subtraction to suppressing the activation-induced dominant singular direction. Without this insight, prior methods remove the mean fully. SGMS identifies this connection and enables a lightweight, partial suppression.

A different line of work employs covariance-based preconditioners, most notably AdaBK (Yong et al., 2023), together with related methods (Gupta et al., 2018; Yun et al., 2019; Grosse & Martens, 2016; Yong & Zhang, 2022; Ma et al., 2025). These approaches explicitly decorrelate weight update directions by estimating second-order statistics and applying matrix transformations. Geometrically, such decorrelation can also be interpreted as amplifying underutilized directions, which aligns with the motivation of SGMS. However, they require maintaining covariance matrices and computing inverse square roots via matrix decompositions or iterative methods, which incur substantial computational and memory overhead. In contrast, SGMS achieves a similar rebalancing effect with only a single mean subtraction, providing a practical lightweight alternative to covariance-based preconditioners.

A related discussion of dominant-direction suppression appears in early work (LeCun et al., 1991). Their analysis shows that when inputs or hidden activations are zero-centered (e.g., when using symmetric activations such as `tanh` instead of Sigmoid), the dominant singular direction becomes less pronounced. Realizing such centering requires modifying the forward pass by altering activation functions or explicitly centering activations. Such forward-pass modifications are not appreciated with modern normalized ReLU/GeLU architectures, where the computational graph and initialization schemes are designed around nonnegative activations. In contrast, SGMS reproduces a similar dominant-direction suppression without any forward-pass changes, using only a lightweight mean-subtraction in backward-pass.

These strands of work remained separate: dominant-direction suppression and gradient centering had been discussed independently, but not connected. Our analysis shows that, under modern ReLU-like activations, mean subtraction closely aligns with suppressing the dominant singular direction, linking these ideas into a unified geometric picture. This connection directly motivates SGMS as a lightweight and principled variant.

## 3 METHODOLOGY

This section is organized as follows. Section 3.1 discusses how LoRA motivates our proposed method. Section 3.2 analyzes the gradient structure of linear layers, motivating the need for rebalancing. Section 3.3 describes the details of SGMS. Section 3.4 discusses the limitations of SGMS.

### 3.1 GRADIENTS OF LINEAR LAYERS LIE IN A LOW-DIMENSIONAL SUBSPACE

It is well known that the training dynamics of DNNs evolve in a low-dimensional subspace relative to the full parameter space. For example, Hu et al. (2022) introduced Low-Rank Adaptation (LoRA), which parameterizes the update of a pre-trained weight matrix $W_0 \in \mathbb{R}^{k \times d}$ as

$$W = W_0 + \Delta W, \quad \Delta W = BA, \tag{1}$$

with $A \in \mathbb{R}^{r \times d}$ and $B \in \mathbb{R}^{k \times r}$, where $r \ll \min(d, k)$. By training only $A$ and $B$ while keeping $W_0$ fixed, LoRA achieves parameter-efficient fine-tuning.

This example highlights that effective updates can be confined to a low-rank subspace without sacrificing accuracy. Consequently, even conventional full-parameter training inherently exploits only a low-dimensional subspace, leaving much of the complementary subspace underutilized. SGMS is motivated by this observation: rather than restricting updates to the dominant span, it explicitly amplifies underutilized directions to unlock additional capacity of trained models.

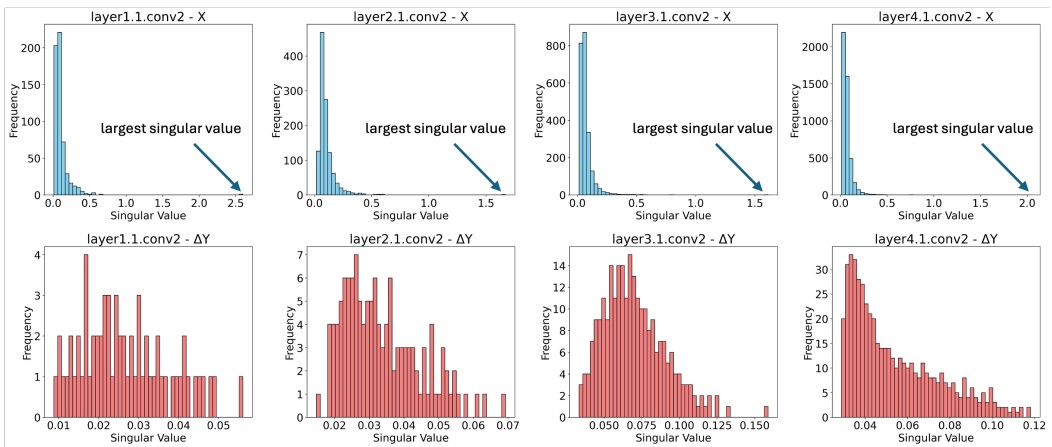

Figure 1: Singular value spectra in several layers of ResNet-18 at random initialization, computed over 40 mini-batches of size 256. Top: histograms of $X$; bottom: those of $\Delta Y$.

## 3.2 ANALYSIS ON GRADIENTS

We now turn to standard full-parameter training regimes, independent of LoRA, to analyze why weight updates naturally concentrate in a low-rank subspace.

Consider a linear transformation

$$Y = XW, \tag{2}$$

with input $X \in \mathbb{R}^{n \times k}$, weights $W \in \mathbb{R}^{k \times d}$, and output $Y \in \mathbb{R}^{n \times d}$. The gradient of the loss with respect to $W$ is

$$\Delta W = \tfrac{1}{n} X^\top \Delta Y, \tag{3}$$

where $\Delta Y \in \mathbb{R}^{n \times d}$ is the backpropagated gradient. The prefactor $1/n$ results from averaging over the input samples and is omitted hereafter, as it does not affect the following rank analysis. From basic linear algebra it follows that

$$\mathrm{rank}(\Delta W) \leq \min(\mathrm{rank}(X), \mathrm{rank}(\Delta Y)), \tag{4}$$

so the rank of $\Delta W$ is always bounded by those of $X$ and $\Delta Y$. In particular, if either $X$ or $\Delta Y$ is degenerate, the weight updates are necessarily confined to a lower-dimensional subspace.

To examine this phenomenon empirically, we computed SVDs of $X$ and $\Delta Y$ at different layers of ResNet-18. The data were collected at random initialization using 10,240 training images from ImageNet-1K (40 mini-batches of size 256). As shown in Figure 1, both $X$ and $\Delta Y$ exhibit skewed spectra, but the skewness is far more pronounced for $X$. In particular, the largest singular value of $X$ is much larger than the rest, whereas $\Delta Y$ is comparatively balanced. We observed the same behavior across other architectures under nonnegative activations such as ReLU. These results indicate that $X$ primarily drives the low-rank structure of $\Delta W$.

When $X$ lies in such a low-rank subspace, $\Delta W$ is effectively restricted to the span of $X$, and directions associated with small singular values contribute negligibly. A straightforward countermeasure is to precondition $\Delta W$ to cancel this skewness. Let the SVD of $X$ be

$$X = U\Sigma V^\top, \quad \Sigma = \mathrm{diag}(\sigma_1, \ldots, \sigma_m), \tag{5}$$

where $m$ is the number of nonzero singular values. Preconditioning with the inverse square root of the Gram matrix $X^\top X$ yields

$$\Delta \bar{W} = \left(X^\top X\right)^{-\frac{1}{2}} \Delta W = VU^\top \Delta Y, \tag{6}$$

which effectively cancels the influence of the dominant singular values of $X$. This procedure corresponds to one component of the full preconditioning performed in AdaBK (Yong et al., 2023), which additionally accounts for $\Delta Y$. While effective, such covariance-based preconditioners require computing SVDs (or iterative alternatives) and matrix multiplications during training, leading to significant computational overhead. This observation motivates the search for lightweight approximations that retain the benefit of rebalancing without the cost of explicit SVD.

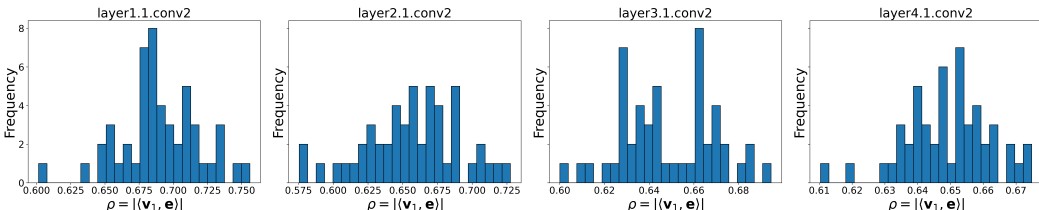

Figure 2: Histogram of $\rho = |\langle \boldsymbol{v}_1, \boldsymbol{e} \rangle|$ across layers, using ResNet-18. Even values around 0.6 indicate strong alignment in high-dimensional space, where random vectors would otherwise be nearly orthogonal.

### 3.3 SCALED GRADIENT MEAN SUBTRACTION

Before introducing our method, we emphasize that we do not solve Eq. (6) or approximate its inverse square root. Our goal is instead to obtain a similar whitening-like effect—reducing the dominance of the largest singular direction—without the heavy computation required by Eq. (6). This becomes feasible when $X$ is nonnegative, where the leading singular direction can be well-approximated by the uniform vector.

Our proposed method, Scaled Gradient Mean Subtraction (SGMS), leverages this property. It smooths the singular value spectrum by subtracting a scaled column mean from $\Delta W$, requiring minimal additional computation. Although SGMS does not fully match the effectiveness of the SVD-based approach discussed in Section 3.2, the trade-off yields a method that is extremely lightweight while still effective in practice.

As observed in Figure 1, the largest singular value of $X$ dominates the gradient updates. Flattening the entire spectrum would counter this imbalance, but requires costly SVD. In practice, suppressing only the dominant direction should already be effective, which simultaneously amplifies the relative contribution of other directions and keeps the method lightweight. This efficiency–effectiveness trade-off motivates the design of SGMS.

We can express $X$ in its singular value expansion as

$$X = \sum_{i=1}^{m} \sigma_i \boldsymbol{u}_i \boldsymbol{v}_i^\top, \tag{7}$$

with singular values ordered as $\sigma_1 \geq \sigma_2 \geq \cdots \geq \sigma_m \geq 0$. Although $X$ is in principle full rank ($m = k$), many singular values are close to zero, so it is effectively degenerate. Rather than flattening the entire spectrum, we attenuate only the largest singular value $\sigma_1$. Specifically, we introduce

$$P = I - \beta \boldsymbol{v}_1 \boldsymbol{v}_1^\top, \tag{8}$$

where $I$ is the identity and $\beta \in [0, 1]$ controls the suppression. Preconditioning $\Delta W$ with $P$ yields

$$\Delta \tilde{W} = P \Delta W = \left( X - \beta X \boldsymbol{v}_1 \boldsymbol{v}_1^\top \right)^\top \Delta Y, \tag{9}$$

which weakens the component of $X$ along $\boldsymbol{v}_1$, thereby amplifying the relative contributions of other directions. Hence it suffices to approximate the leading right singular vector $\boldsymbol{v}_1$, which could in principle be obtained by power iteration on $X^\top X$. However, such iterations during training would incur significant computational overhead, which is undesirable in practice.

Fortunately, there is a simple and practically effective approximation for $\boldsymbol{v}_1$. When the elements of $X$ are nonnegative, $X^\top X$ becomes a nonnegative symmetric matrix, and by the Perron–Frobenius theorem its leading eigenvector has nonnegative entries. This does not imply that $\boldsymbol{v}_1$ is exactly the uniform direction $\boldsymbol{e}$, but it ensures that $\boldsymbol{v}_1$ lies in the positive orthant and therefore has a strictly positive correlation with $\boldsymbol{e}$. If the columns of $X$ exhibit broadly similar activation statistics under ReLU-like nonlinearities, this makes $\boldsymbol{v}_1$ close to $\boldsymbol{e}$. These observations motivate using $\boldsymbol{e}$ as a lightweight approximation of the dominant direction for SGMS. A mathematical discussion of the robustness of this approximation under perturbations is provided in Appendix H.

Formally, we define

$$\boldsymbol{e} = \frac{1}{\sqrt{k}} [1, 1, \dots, 1]^\top \in \mathbb{R}^k, \qquad \rho = \left| \langle \boldsymbol{v}_1, \boldsymbol{e} \rangle \right|, \tag{10}$$

where $\langle \cdot, \cdot \rangle$ denotes the standard inner product, and $\rho$ quantifies how closely $\boldsymbol{v}_1$ aligns with $\boldsymbol{e}$. Figure 2 shows the empirical distribution of $\rho$ in ResNet-18. The distribution indicates strong alignment between $\boldsymbol{v}_1$ and $\boldsymbol{e}$. Even values around 0.6, which may appear moderate at first glance, imply strong correlation, since in high-dimensional spaces random vectors are nearly orthogonal. This observation suggests that weakening the projection onto $\boldsymbol{e}$, instead of $\boldsymbol{v}_1$, effectively suppresses the dominant direction associated with the largest singular value.

With this approximation, we can now summarize our method. The preconditioner $P$ that originally required $\boldsymbol{v}_1$ can be replaced by a computationally lightweight form using $\boldsymbol{e}$:

$$P' = I - \beta \boldsymbol{e}\boldsymbol{e}^\top. \tag{11}$$

Accordingly, the updates become

$$\Delta \tilde{W}' = P'\Delta W = \Delta W - \beta \boldsymbol{e}\boldsymbol{e}^\top \Delta W. \tag{12}$$

Since $\boldsymbol{e}\boldsymbol{e}^\top$ simply averages the columns of a matrix, this operation amounts to subtracting the scaled column mean from each column of $\Delta W$.

Ideally, the coefficient $\beta$ would be chosen in a statistically principled way, for example by selecting the value that maximizes the spectral entropy of the layer input $X$ after the rank-one correction induced by SGMS. However, computing such a quantity would require estimating singular values or covariance matrices during training, which would introduce an overhead comparable to that of second-order preconditioners such as Shampoo or AdaBK. To keep SGMS lightweight, we simply treat $\beta$ as a hyperparameter and select a fixed value.

Note that SGMS also applies to convolution and attention layers, because their core operations can be expressed as matrix multiplications; see Appendix F for details.

**Relation to GC.**  When $\beta = 1$, SGMS coincides exactly with Gradient Centralization (GC) (Yong et al., 2020). We denote this setting simply as GC in our experiments, with further details and explicit formulas given in Appendix D.

**Relation to AdaBK.**  SGMS can also be related to AdaBK (Yong et al., 2023), which preconditions updates based on covariance structure. A more detailed discussion is provided in Appendix E.

**Optional rescaling.**  An additional step is to rescale $\Delta \tilde{W}'$ so that its Frobenius norm matches that of the original update. Since adaptive optimizers already normalize updates in practice, this step has negligible effect; we defer details and empirical results to Appendix C.4.

### 3.4 LIMITATION

SGMS assumes that the input $X$ to each linear layer is nonnegative, which motivates our approximation of the leading singular vector by the uniform direction $\boldsymbol{e}$ (Section 3.3). This holds for widely used activations such as ReLU.

SGMS also remains effective with activations that can take negative values but are biased toward positive outputs, such as SiLU/Swish (Ramachandran et al., 2017) or GeLU (Hendrycks & Gimpel, 2016), which are widely used in Transformer models (Dosovitskiy et al., 2021; Liu et al., 2021; Touvron et al., 2021b; Wang et al., 2021). In support of this, we provide additional empirical evidence in Appendix I, where we show that GeLU activations in a pretrained RoBERTa encoder still yield a leading singular direction strongly aligned with $\boldsymbol{e}$. For zero-centered activations (e.g., tanh, linear), the benefit is weaker; however, since their mean is already near zero, subtracting it has little effect and is effectively neutral.

These activation types can be summarized as follows:

- *Strictly nonnegative (theoretically valid):* ReLU, ReLU6, Sigmoid, Softplus.
- *Non-strictly nonnegative (empirically effective):* SiLU / Swish, GeLU.
- *Zero-centered (neutral effect):* Tanh, Linear.

In summary, SGMS is theoretically justified under nonnegativity, empirically effective beyond this regime, and essentially neutral in zero-centered settings.

Table 1: Top-1 accuracy (%, mean ± std over 5 runs) with training cost on CIFAR-100 (Adam). "Acc. gain" indicates improvement relative to the Vanilla baseline. Time is per epoch (s, measured on an RTX 4060 GPU), Mem is peak GPU memory (GiB). Values in parentheses for Time/Mem are normalized to Vanilla (=1.00). "SGMS, $\beta = 1$ (GC)" denotes pure GC (mean subtraction only; no rescaling). No rescaling is used under Adam/AdamW for any $\beta$.

| Model | Method | Top-1 Acc. (%) | Acc. gain | Time (s) | Mem (GiB) |
|---|---|---|---|---|---|
| ResNet-32 | Vanilla | 69.21 ± 0.38 | – | 4.39 | 0.35 |
| | SGMS, $\beta = 0.9$ | 69.67 ± 0.26 | +0.46 | 4.46 (×1.02) | 0.35 (×1.00) |
| | SGMS, $\beta = 1$ (GC) | 69.50 ± 0.27 | +0.29 | 4.42 (×1.01) | 0.35 (×1.00) |
| | AdaBK | 70.57 ± 0.21 | +1.36 | 4.76 (×1.08) | 0.83 (×2.32) |
| ResNet-56 | Vanilla | 69.87 ± 0.37 | – | 7.25 | 0.55 |
| | SGMS, $\beta = 0.9$ | 70.69 ± 0.42 | +0.82 | 7.31 (×1.01) | 0.55 (×1.00) |
| | SGMS, $\beta = 1$ (GC) | 70.42 ± 0.23 | +0.55 | 7.26 (×1.00) | 0.55 (×1.00) |
| | AdaBK | 72.41 ± 0.20 | +2.54 | 7.90 (×1.09) | 1.05 (×1.93) |

## 4 EXPERIMENTS

We evaluate SGMS across both vision and language modeling tasks. For image classification, we consider CIFAR-100 (Krizhevsky, 2009) as a lightweight benchmark and ImageNet-1K (Deng et al., 2009) as a large-scale benchmark, using ResNets (He et al., 2016) and DeiT (Touvron et al., 2021a) as representative architectures. For language modeling, we additionally conduct experiments on a compact GPT-style model trained from scratch on a 1M-token subset of OpenWebText (Section 4.3).

Unless otherwise noted, the SGMS scaling factor is set to $\beta = 0.9$; an ablation study on the choice of $\beta$ is provided in Appendix C.3. Detailed implementation settings are summarized in Appendix A. Additional CIFAR-10/100 results appear in Appendix B, and extended ablations together with further discussions are presented in Appendix C.

### 4.1 RESULTS ON CIFAR-100 WITH ADAM OPTIMIZER

Table 1 reports CIFAR-100 results with Adam for two typical mid-sized ResNets (32 and 56); full results for all ResNets are in Appendix B. SGMS with $\beta = 0.9$ consistently improves accuracy over Vanilla, with only minor runtime overhead and no additional memory cost. SGMS with $\beta = 1$ (GC) also improves upon Vanilla, but is generally slightly worse than $\beta = 0.9$. Occasionally GC outperforms in individual runs (Appendix B), yet the overall trend favors partial suppression. This matches our interpretation: fully removing the mean-gradient direction ($\beta = 1$, GC) can discard useful information in that direction, whereas partial suppression ($0 < \beta < 1$, SGMS) preserves it while reducing its dominance.

AdaBK yields the largest improvements (e.g., +2.5 points on ResNet-56), but incurs nearly 2× higher memory usage and noticeable runtime overhead. SGMS therefore provides a more favorable accuracy–efficiency trade-off as a lightweight alternative to costly preconditioners.

The observed runtime increase for SGMS (up to 3%) is much larger than the theoretical estimate (e.g., 0.0026% for ResNet-20). This discrepancy likely stems from implementation details, as the rest of the training loop uses highly optimized native kernels. With further optimization, the overhead of SGMS should become negligible (Appendix G).

### 4.2 RESULTS ON IMAGENET-1K

Having established the effectiveness of SGMS on CIFAR-100, we next evaluate its performance on the large-scale ImageNet-1K dataset. For ResNet-18 and ResNet-50 (He et al., 2016), we follow the conventional 90-epoch training recipe. For DeiT-Small (Touvron et al., 2021a), we adopt the standard 300-epoch ImageNet training schedule with AdamW.

Table 2 and Figures 3–4 summarize the ImageNet-1K results. Across ResNet-18, ResNet-50, and DeiT-Small, SGMS consistently improves over the Vanilla baseline while incurring negligible over-

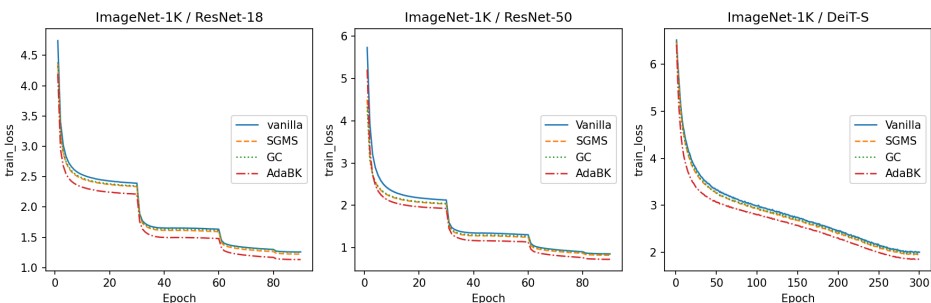

Figure 3: Training loss vs. epoch on ImageNet (mean ± std. over 3 runs). Each panel compares Vanilla, SGMS, GC, and AdaBK, with shaded regions indicating the standard deviation across runs. *AdaBK (RB)* denotes AdaBK trained with a reduced batch size due to memory constraints.

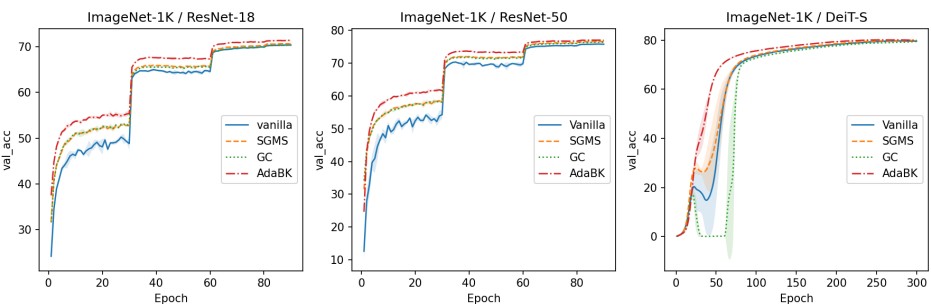

Figure 4: Validation accuracy vs. epoch on ImageNet (mean ± std. over 3 runs). Each panel compares Vanilla, SGMS, GC, and AdaBK, with shaded regions indicating the standard deviation across runs. *AdaBK (RB)* denotes AdaBK trained with a reduced batch size due to memory constraints.

head in both runtime and memory. For example, SGMS yields gains of +0.23 points on ResNet-18 and +0.83 points on ResNet-50 with essentially identical training cost. On DeiT-Small, SGMS also produces a clear improvement (+0.24 points), demonstrating that the method transfers to Transformer architectures as well. GC generally exhibits trends similar to SGMS, whereas AdaBK achieves larger gains (e.g., +1.01 on ResNet-18 and +0.37 on DeiT-Small) at substantially higher memory cost.

We further analyze the optimization dynamics using the training loss curves shown in Figure 3. Across all architectures, SGMS reduces the training loss more rapidly than the Vanilla baseline and exhibits stable convergence, confirming that the method improves the underlying optimization process. GC shows broadly similar behavior, whereas AdaBK achieves faster loss reduction at the cost of a substantially higher memory footprint. A short temporary dip in validation accuracy appears around the learning-rate decay points (Figure 4), a phenomenon that also occurs for Vanilla and GC; this fluctuation is common in ImageNet training and does not indicate instability in the optimization, as the training loss continues to decrease monotonically during this period.

## 4.3 RESULTS ON LANGUAGE MODELS

To evaluate SGMS in an autoregressive language modeling setting, we conducted experiments using a minimal GPT-style model based on the `nano_gpt.py` implementation from whatdhack (2024).Apart from adding SGMS, the only modification to the original code was to introduce an explicit split into train/validation/test sets. We used a subset of OpenWebText containing one million tokens, which we partitioned into training, validation, and test splits in an 8:1:1 ratio.

We adopt a compact GPT architecture (4 layers, 6 heads, 384-dim embeddings, context length 1024), closely following the original `nano_gpt.py` implementation with dropout disabled. We use gradient accumulation with 4 steps, which corresponds to an effective batch size of roughly 250K tokens.

Table 2: Top-1 accuracy (%, mean ± std over 3 runs) with training cost on ImageNet-1K. "Acc. gain" indicates the accuracy improvement relative to the Vanilla baseline. Time is per epoch (measured on an RTX 3090 GPU), Mem is peak GPU memory. Values in parentheses for Time/Mem are normalized to Vanilla (=1.00); N.A. indicates cases where batch size differs. "RB" denotes results obtained with a reduced batch size due to memory constraints. GC has the same computational form as SGMS (per-layer mean subtraction) and therefore the same time and memory cost; we omit these redundant numbers for clarity. ResNets are trained with SGD, and DeiT-Small is trained with AdamW.

| Model | Method | Acc. (%) | Acc. gain | Time (s/epoch) | Mem (GiB) |
|---|---|---|---|---|---|
| ResNet-18 | Vanilla | $70.35 \pm 0.01$ | – | 544.8 | 3.25 |
| | SGMS | $70.58 \pm 0.15$ | +0.23 | 542.9 ($\times 1.00$) | 3.25 ($\times 1.00$) |
| | GC | $70.34 \pm 0.08$ | -0.01 | – | – |
| | AdaBK | $71.36 \pm 0.11$ | +1.01 | 590.9 ($\times 1.08$) | 10.11 ($\times 3.11$) |
| ResNet-50 | Vanilla | $75.71 \pm 0.01$ | – | 1413.4 | 11.68 |
| | SGMS | $76.54 \pm 0.03$ | +0.83 | 1416.3 ($\times 1.00$) | 11.68 ($\times 1.00$) |
| | GC | $76.30 \pm 0.11$ | +0.59 | – | – |
| | AdaBK | | | Out of memory | |
| | AdaBK (RB) | $76.89 \pm 0.11$ | +1.18 | 1599.8 (N.A.) | 9.34 (N.A.) |
| DeiT-Small | Vanilla | $79.58 \pm 0.25$ | – | 1160.6 | 8.81 |
| | SGMS | $79.82 \pm 0.15$ | +0.24 | 1162.3 ($\times 1.00$) | 8.81 ($\times 1.00$) |
| | GC | $79.48 \pm 0.07$ | -0.10 | – | – |
| | AdaBK | $79.95 \pm 0.08$ | +0.37 | 1177.4 ($\times 1.01$) | 15.23 ($\times 1.73$) |

Table 3: Test-set loss of nanoGPT for different SGMS strengths $\beta$.

| $\beta$ | 0.0 (vanilla) | 0.1 | 0.3 | 0.5 | 0.7 | 0.9 | 1.0 (GC) |
|---|---|---|---|---|---|---|---|
| Test loss | 1.0423 | 1.0394 | 1.0415 | 1.0355 | **1.0321** | 1.0326 | 1.0374 |

Accordingly, we scale the learning rate to $1.2 \times 10^{-3}$ (from $6 \times 10^{-4}$ in the original script) following the standard $\sqrt{\text{batch\_size}}$ rule.

We vary the SGMS strength via $\beta \in \{0.0 \text{ (vanilla)}, 0.1, 0.3, 0.5, 0.7, 0.9, 1.0 \text{ (GC)}\}$. The final test losses are summarized in Table 3. Although the improvements are modest, SGMS shows a consistent trend of lower loss compared to the vanilla baseline across all values of $\beta$, with $\beta = 0.7$ achieving the best overall performance. The full training and validation loss trajectories are shown in Fig. 5, exhibiting the same pattern of steady but reliable improvement across the training process. These results indicate that SGMS provides a small yet stable benefit in GPT-style autoregressive language modeling without introducing instability or requiring architectural modifications.

We additionally evaluated SGMS on GLUE fine-tuning tasks, and the results are provided in Appendix I.1.

# 5 ADDITIONAL ANALYSES AND ABLATIONS

We provide several supplementary analyses supporting the behavior of SGMS.

First, spectral measurements on ResNet-50/ImageNet show that SGMS increases both stable rank and entropy-based rank, confirming suppression of the dominant singular direction and redistribution of spectral mass toward smaller ones (App. C.1).

Second, activation-function experiments indicate that SGMS is most effective under ReLU and also works under positively biased nonlinearities such as SiLU and GeLU; zero-centered activations (e.g., Tanh) show negligible effects, in line with our nonnegativity rationale (App. C.2).

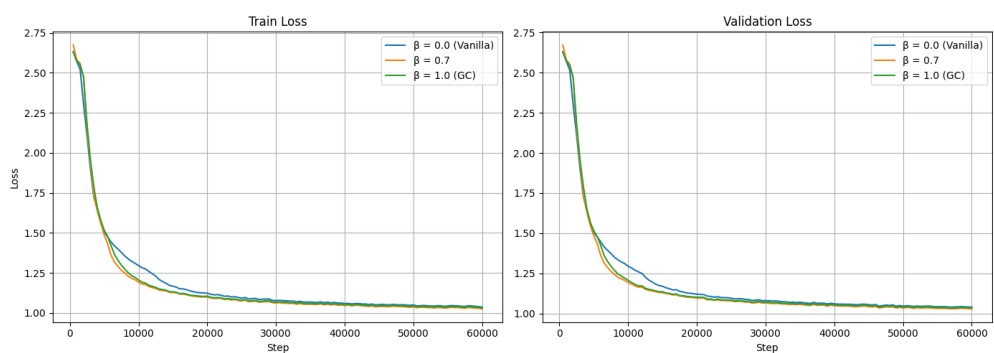

Figure 5: Training and validation loss curves for nanoGPT with varying SGMS strengths $\beta$.

Third, a CIFAR-100 sweep finds that partial suppression performs best: $\beta \in [0.8, 0.9]$ consistently outperforms both $\beta = 0$ and $\beta = 1$, so we use $\beta = 0.9$ as the default in all experiments (App. C.3).

Finally, an optional Frobenius-norm rescaling yields small gains with SGD but not with Adam; we therefore apply it only in SGD settings (App. C.4).

These results collectively support the theoretical motivation in Section 3: SGMS is most effective when activations are positively biased and when the dominant direction is only partially suppressed.

## 6 CONCLUSION

We introduced *Scaled Gradient Mean Subtraction (SGMS)*, a lightweight method for improving neural network training by *amplifying underutilized gradient directions*. SGMS is motivated by the empirical observation that gradients in DNNs often concentrate in a low-dimensional subspace, leaving many other directions underutilized. Our method suppresses the dominant gradient direction through a simple mean-subtraction step, optionally followed by rescaling, thereby amplifying the relative contributions of the other directions. Experiments on CIFAR and ImageNet show that SGMS improves accuracy with minimal overhead, yielding substantial gains on CNNs and small but consistent improvements on Transformers, while remaining simple and easy to deploy in existing architectures. SGMS is also optimizer-agnostic, making it broadly applicable across diverse training pipelines. An important special case is $\beta = 1$, corresponding to *Gradient Centralization (GC)*; while GC improves over vanilla training, partial suppression with $\beta < 1$ often yields larger gains by reducing—but not eliminating—the dominant direction.

**LLM Usage Statement.** See Appendix J for our LLM usage statement.

## REPRODUCIBILITY STATEMENT

All datasets used in this work (CIFAR-10, CIFAR-100, and ImageNet-1K) are publicly available. We provide detailed descriptions of model architectures, hyperparameters, and training schedules in the main text and Appendix A. Random seeds were fixed to facilitate reproducibility. The source code used in our experiments will be released upon acceptance.

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

## A  EXPERIMENTAL SETUP DETAILS

We summarize the detailed training configurations used in our experiments.

**ResNets and plain networks on CIFAR-10/100.**    We train for 200 epochs (100 epochs for ablation studies) with a batch size of 256, using standard data augmentations including random cropping with 4-pixel padding and random horizontal flipping. For ResNets and Plain networks, we adopt two types of optimizers:

- **SGD:** initial learning rate of 0.1, momentum of 0.9, weight decay of $5 \times 10^{-4}$, and learning rate schedule of multi-step decay (factor 0.1) at epochs 100 and 150.
- **Adam:** initial learning rate of $3 \times 10^{-3}$, weight decay of $5 \times 10^{-4}$, $(\beta_1, \beta_2) = (0.9, 0.999)$, $\epsilon = 10^{-8}$, and learning rate schedule of cosine annealing, with a linear warmup of 5 epochs.

**ImageNet-1K.**    We train on the ImageNet-1K dataset with a standard input resolution of $224 \times 224$. Training runs differ depending on the model family:

- **ResNets:** trained for 90 epochs with a batch size of 256. The learning rate is initialized at 0.1, multi-step schedule at epochs 30, 60, and 80, with weight decay of $10^{-4}$. Data augmentation includes random resized crops, random horizontal flipping, and normalization with ImageNet mean and standard deviation.
- **Transformer-based models (DeiT):** trained for 300 epochs with a batch size of 1024 (a batch size of 256 with gradient accumulation of 4 steps). The learning rate is initialized at $5 \times 10^{-4}$, decayed by cosine annealing, with weight decay of 0.05. In addition to the standard augmentations, we follow Touvron et al. (2021a) and apply RandAugment, Mixup, and CutMix. We also use label smoothing and stochastic depth with drop-path rate 0.1, together with a linear warmup of 5 epochs, as commonly adopted in Transformer training.

**Reporting.**    Results are averaged over 5 independent runs on CIFAR datasets and 3 runs on ImageNet. We report the mean top-1 accuracy at the final epoch as well as training time and memory cost.

## B  ADDITIONAL RESULTS ON CIFAR-10 AND CIFAR-100

Tables 4 and 5 report results on CIFAR-100 with SGD and Adam optimizers, while Tables 6 and 7 summarize the corresponding results on CIFAR-10. Across both datasets and optimizers, the trends are consistent with those presented in the main text: SGMS provides accuracy improvements over Vanilla training while incurring only minimal overhead.

**Runtime overhead.**    Regarding training time, SGMS shows a small (up to 3%) increase across models, which is much larger than the theoretical estimate (e.g., 0.0026% for ResNet-20; see Appendix G). We attribute this discrepancy to our own implementation, whereas the rest of the training loop uses highly optimized native kernels. With a more optimized implementation, the overhead of SGMS would likely become negligible.

## C  ANALYSES AND ABLATIONS

We conduct a series of ablation studies and analyses to better understand the behavior of SGMS. Unless otherwise noted, all experiments in this section are run on CIFAR-100 with a shortened 100-epoch schedule to reduce training cost.

### C.1  SPECTRAL ANALYSIS OF TRAINED WEIGHTS

To further validate our motivation, we examined the singular value spectra of trained weight matrices in ResNet-50.

Table 4: Top-1 accuracy (%, mean ± std over 5 runs) with training cost on CIFAR-100 (SGD). "Acc. gain" indicates improvement relative to the Vanilla baseline. Time is per epoch (s, measured on an RTX 4060 GPU), Mem is peak GPU memory (GiB). Values in parentheses for Time/Mem are normalized to Vanilla (=1.00). "SGMS, $\beta = 1$ (GC)" denotes pure GC (mean subtraction only; no rescaling). For $\beta < 1$, we apply Frobenius-norm rescaling under SGD, while no rescaling is used under Adam/AdamW.

| Model | Method | Top-1 Acc. (%) | Acc. gain | Time (s) | Mem (GiB) |
|---|---|---|---|---|---|
| Plain-20 | Vanilla | 66.19 ± 0.35 | – | 2.78 | 0.28 |
| | SGMS, $\beta = 0.9$ | 66.67 ± 0.20 | +0.48 | 2.81 (×1.01) | 0.28 (×1.00) |
| | SGMS, $\beta = 1$ (GC) | 66.44 ± 0.48 | +0.25 | 2.81 (×1.01) | 0.28 (×1.00) |
| | AdaBK | 67.10 ± 0.25 | +0.91 | 2.97 (×1.07) | 0.69 (×2.49) |
| ResNet-20 | Vanilla | 67.94 ± 0.26 | – | 2.94 | 0.29 |
| | SGMS, $\beta = 0.9$ | 68.06 ± 0.34 | +0.12 | 2.94 (×1.00) | 0.29 (×1.00) |
| | SGMS, $\beta = 1$ (GC) | 68.35 ± 0.33 | +0.41 | 2.97 (×1.01) | 0.29 (×1.00) |
| | AdaBK | 68.39 ± 0.28 | +0.45 | 3.16 (×1.07) | 0.69 (×2.36) |
| ResNet-32 | Vanilla | 69.69 ± 0.36 | – | 4.27 | 0.35 |
| | SGMS, $\beta = 0.9$ | 69.85 ± 0.29 | +0.16 | 4.31 (×1.01) | 0.35 (×1.00) |
| | SGMS, $\beta = 1$ (GC) | 69.82 ± 0.18 | +0.13 | 4.26 (×1.00) | 0.35 (×1.00) |
| | AdaBK | 70.53 ± 0.53 | +0.84 | 4.59 (×1.07) | 0.81 (×2.32) |
| ResNet-44 | Vanilla | 70.26 ± 0.18 | – | 5.74 | 0.44 |
| | SGMS, $\beta = 0.9$ | 70.86 ± 0.43 | +0.60 | 5.87 (×1.02) | 0.44 (×1.00) |
| | SGMS, $\beta = 1$ (GC) | 70.66 ± 0.45 | +0.40 | 5.74 (×1.00) | 0.44 (×1.00) |
| | AdaBK | 71.77 ± 0.24 | +1.51 | 6.18 (×1.08) | 0.93 (×2.11) |
| ResNet-56 | Vanilla | 70.51 ± 0.40 | – | 7.17 | 0.54 |
| | SGMS, $\beta = 0.9$ | 71.86 ± 0.46 | +1.35 | 7.38 (×1.03) | 0.54 (×1.00) |
| | SGMS, $\beta = 1$ (GC) | 71.38 ± 0.41 | +0.87 | 7.21 (×1.01) | 0.54 (×1.00) |
| | AdaBK | 72.46 ± 0.29 | +1.95 | 7.74 (×1.08) | 1.05 (×1.93) |

**Definition of stable rank.** Given a weight matrix $W$ with singular values $\sigma_1 \geq \sigma_2 \geq \cdots \geq \sigma_r > 0$, the *stable rank* (Vershynin, 2018) is defined as

$$\mathrm{srank}(W) \;=\; \frac{\|W\|_{\mathrm{F}}^2}{\|W\|_2^2} \;=\; \frac{\sum_{i=1}^r \sigma_i^2}{\sigma_1^2}, \tag{13}$$

where $\|\cdot\|_2$ represents the spectral norm. It satisfies $1 \leq \mathrm{srank}(W) \leq r$, equals 1 when $W$ is effectively rank-one (dominated by its top singular value), and equals $r$ when all singular values are equal. Hence, an increase in stable rank indicates that spectral energy is more evenly distributed across singular directions.

**Definition of entropy-based rank.** In addition to the stable rank, we also report an *entropy-based rank*, sometimes referred to as an "effective rank" in the literature (Roy & Vetterli, 2007). It is defined as

$$r_{\mathrm{ent}}(W) \;=\; \exp\left(-\sum_{i=1}^r p_i \log p_i\right), \quad p_i = \frac{\sigma_i^2}{\sum_{j=1}^r \sigma_j^2}. \tag{14}$$

This measure corresponds to the entropy of the normalized singular value distribution, and increases when the spectrum becomes more balanced. Note that another notion of effective rank, defined as $\mathrm{tr}(\Sigma)/\|\Sigma\|$ for a covariance matrix (Vershynin, 2018), is also common; in this work we focus on the entropy-based definition.

**Results.** Figure 6 compares the stable rank (**left**) and entropy-based rank (**right**) of each ResNet-50 layer trained with vanilla SGD (x-axis) and SGD+SGMS ($\beta = 0.9$, y-axis). In both metrics, most points lie above the diagonal, indicating that SGMS tends to increase the spectral dispersion across layers. This supports our claim that SGMS mitigates the dominance of the largest singular direction and amplifies underutilized directions, thereby promoting a more balanced use of multiple dimensions.

Table 5: Top-1 accuracy (%, mean ± std over 5 runs) with training cost on CIFAR-100 (Adam). "Acc. gain" indicates improvement relative to the Vanilla baseline. Time is per epoch (s, measured on an RTX 4060 GPU), Mem is peak GPU memory (GiB). Values in parentheses for Time/Mem are normalized to Vanilla (=1.00). "SGMS, $\beta = 1$ (GC)" denotes pure GC (mean subtraction only; no rescaling). No rescaling is used under Adam/AdamW for any $\beta$.

| Model | Method | Top-1 Acc. (%) | Acc. gain | Time (s) | Mem (GiB) |
|---|---|---|---|---|---|
| Plain-20 | Vanilla | 65.04 ± 0.15 | – | 2.85 | 0.28 |
| | SGMS, $\beta = 0.9$ | 65.93 ± 0.54 | +0.89 | 2.86 (×1.00) | 0.28 (×1.00) |
| | SGMS, $\beta = 1$ (GC) | 65.48 ± 0.39 | +0.44 | 2.85 (×1.00) | 0.28 (×1.00) |
| | AdaBK | 65.92 ± 0.33 | +0.88 | 3.05 (×1.07) | 0.71 (×2.49) |
| ResNet-20 | Vanilla | 67.32 ± 0.30 | – | 2.99 | 0.29 |
| | SGMS, $\beta = 0.9$ | 67.65 ± 0.27 | +0.33 | 3.02 (×1.01) | 0.29 (×1.00) |
| | SGMS, $\beta = 1$ (GC) | 67.67 ± 0.32 | +0.35 | 3.02 (×1.01) | 0.29 (×1.00) |
| | AdaBK | 67.58 ± 0.35 | +0.26 | 3.22 (×1.08) | 0.71 (×2.36) |
| ResNet-32 | Vanilla | 69.21 ± 0.38 | – | 4.39 | 0.35 |
| | SGMS, $\beta = 0.9$ | 69.67 ± 0.26 | +0.46 | 4.46 (×1.02) | 0.35 (×1.00) |
| | SGMS, $\beta = 1$ (GC) | 69.50 ± 0.27 | +0.29 | 4.42 (×1.01) | 0.35 (×1.00) |
| | AdaBK | 70.57 ± 0.21 | +1.36 | 4.76 (×1.08) | 0.83 (×2.32) |
| ResNet-44 | Vanilla | 69.51 ± 0.19 | – | 5.86 | 0.44 |
| | SGMS, $\beta = 0.9$ | 70.29 ± 0.09 | +0.78 | 5.91 (×1.01) | 0.44 (×1.00) |
| | SGMS, $\beta = 1$ (GC) | 70.19 ± 0.16 | +0.68 | 5.91 (×1.01) | 0.44 (×1.00) |
| | AdaBK | 71.91 ± 0.24 | +2.40 | 6.33 (×1.08) | 0.95 (×2.11) |
| ResNet-56 | Vanilla | 69.87 ± 0.37 | – | 7.25 | 0.55 |
| | SGMS, $\beta = 0.9$ | 70.69 ± 0.42 | +0.82 | 7.31 (×1.01) | 0.55 (×1.00) |
| | SGMS, $\beta = 1$ (GC) | 70.42 ± 0.23 | +0.55 | 7.26 (×1.00) | 0.55 (×1.00) |
| | AdaBK | 72.41 ± 0.20 | +2.54 | 7.90 (×1.09) | 1.05 (×1.93) |

## C.2   EFFECT OF ACTIVATION FUNCTIONS

SGMS assumes that the output $X$ of each layer is nonnegative, so that the dominant singular vector can be well approximated by the uniform direction (Section 3.4). To verify this assumption, we tested SGMS with different activation functions. With ReLU, which strictly satisfies the non-negativity condition, SGMS improved accuracy. With SiLU and GeLU, which do not strictly satisfy the condition but still bias activations toward positive values, SGMS also brought considerable gains. In contrast, with Tanh, which violates the condition entirely by producing zero-centered outputs, SGMS failed to provide improvements. These results align with our theoretical analysis and support the core idea of SGMS: when activations are nonnegative, a simple mean subtraction is sufficient to suppress the dominant direction and amplify underutilized ones.

## C.3   ABLATION ON THE SCALING FACTOR

To quickly examine the sensitivity to the scaling factor $\beta$, we trained ResNet-20 and ResNet-56 on CIFAR-100. Table 9 summarizes the results. Both models exhibit stable improvements over the vanilla baseline for a wide range of $\beta$ values. The accuracy peaks around $\beta = 0.8$–$0.9$, indicating that relatively strong suppression of the dominant direction is most effective in practice. Unless otherwise noted, we therefore set $\beta = 0.9$ in all other experiments. Note that all subsequent ablations in this appendix are also conducted with $\beta = 0.9$ as well.

## C.4   ABLATION ON FROBENIUS-NORM RESCALING

An optional step is to rescale $\Delta \tilde{W}'$ so that its Frobenius norm matches that of the original:

$$\Delta \hat{W} = \Delta \tilde{W}' \cdot \frac{\|\Delta W\|_F}{\|\Delta \tilde{W}'\|_F}. \tag{15}$$

Table 6: Top-1 accuracy (%, mean ± std over 5 runs) with training cost on CIFAR-10 (SGD). "Acc. gain" indicates improvement relative to the Vanilla baseline. Time is per epoch (s, measured on an RTX 4060 GPU), Mem is peak GPU memory (GiB). Values in parentheses for Time/Mem are normalized to Vanilla (=1.00). "SGMS, $\beta = 1$ (GC)" denotes pure GC (mean subtraction only; no rescaling). For $\beta < 1$, we apply Frobenius-norm rescaling under SGD, while no rescaling is used under Adam/AdamW.

| Model | Method | Acc. (%) | Acc. gain | Time (s/epoch) | Mem (GiB) |
|---|---|---|---|---|---|
| Plain-20 | Vanilla | 91.30 ± 0.14 | – | 2.79 | 0.27 |
| | SGMS, $\beta = 0.9$ | 91.25 ± 0.22 | –0.05 | 2.79 (×1.00) | 0.27 (×1.00) |
| | SGMS, $\beta = 1$ (GC) | 91.16 ± 0.17 | –0.14 | 2.81 (×1.01) | 0.28 (×1.00) |
| | AdaBK | 91.90 ± 0.25 | +0.60 | 2.98 (×1.07) | 0.71 (×2.49) |
| ResNet-20 | Vanilla | 92.22 ± 0.20 | – | 2.95 | 0.29 |
| | SGMS, $\beta = 0.9$ | 92.16 ± 0.22 | –0.06 | 2.95 (×1.00) | 0.29 (×1.00) |
| | SGMS, $\beta = 1$ (GC) | 92.03 ± 0.33 | –0.19 | 2.96 (×1.00) | 0.29 (×1.00) |
| | AdaBK | 92.57 ± 0.14 | +0.35 | 3.17 (×1.07) | 0.71 (×2.36) |
| ResNet-32 | Vanilla | 92.92 ± 0.27 | – | 4.26 | 0.35 |
| | SGMS, $\beta = 0.9$ | 93.18 ± 0.11 | +0.26 | 4.31 (×1.01) | 0.35 (×1.00) |
| | SGMS, $\beta = 1$ (GC) | 93.03 ± 0.21 | +0.11 | 4.29 (×1.01) | 0.35 (×1.00) |
| | AdaBK | 93.40 ± 0.21 | +0.48 | 4.59 (×1.08) | 0.81 (×2.32) |
| ResNet-44 | Vanilla | 93.11 ± 0.53 | – | 5.73 | 0.44 |
| | SGMS, $\beta = 0.9$ | 93.66 ± 0.18 | +0.55 | 5.84 (×1.02) | 0.44 (×1.00) |
| | SGMS, $\beta = 1$ (GC) | 93.46 ± 0.13 | +0.35 | 5.78 (×1.01) | 0.47 (×1.07) |
| | AdaBK | 93.46 ± 0.07 | +0.35 | 6.21 (×1.08) | 0.93 (×2.11) |
| ResNet-56 | Vanilla | 93.29 ± 0.64 | – | 7.17 | 0.54 |
| | SGMS, $\beta = 0.9$ | 93.67 ± 0.28 | +0.38 | 7.39 (×1.03) | 0.54 (×1.00) |
| | SGMS, $\beta = 1$ (GC) | 93.73 ± 0.08 | +0.44 | 7.23 (×1.01) | 0.56 (×1.04) |
| | AdaBK | 93.90 ± 0.27 | +0.61 | 7.75 (×1.08) | 1.05 (×1.93) |

This normalization preserves the scale of the updates while still suppressing the dominant uniform component. However, when SGMS is combined with adaptive optimizers such as Adam, the rescaling effect is largely absorbed by their per-parameter learning-rate adjustment, and this step can therefore be omitted without loss of effectiveness.

We evaluate the effect of the optional Frobenius-norm rescaling step on CIFAR-100 with ResNet-20 and ResNet-56. As shown in Table 10, rescaling improves the final accuracy for ResNet-20, whereas ResNet-56 performs better without it. This indicates that the empirical benefit of rescaling is not entirely consistent across models. For SGD-based training, rescaling is theoretically aligned with the goal of SGMS, so we apply it in all SGD experiments throughout the paper. For Adam-based training, on the other hand, rescaling is not required since Adam already normalizes updates adaptively.

**Reporting note for tables.** For SGD-based training, we apply Frobenius-norm rescaling to all SGMS rows for theoretical consistency; GC is reported as a separate row corresponding to $\beta = 1$ with no rescaling (pure GC). For Adam-based training, rescaling is omitted since adaptive normalization absorbs its effect.

# D    GC AND ITS RELATION TO SGMS

We define Gradient Centralization (GC) as mean subtraction without rescaling. In our formulation, SGMS with $\beta = 1$ coincides exactly with GC. For $0 \le \beta < 1$, SGMS performs a partial subtraction and may optionally apply Frobenius-norm rescaling.

Table 7: Top-1 accuracy (%, mean ± std over 5 runs) with training cost on CIFAR-10 (Adam). "Acc. gain" indicates improvement relative to the Vanilla baseline. Time is per epoch (s, measured on an RTX 4060 GPU), Mem is peak GPU memory (GiB). Values in parentheses for Time/Mem are normalized to Vanilla (=1.00). "SGMS, $\beta = 1$ (GC)" denotes pure GC (mean subtraction only; no rescaling). No rescaling is used under Adam/AdamW for any $\beta$.

| Model | Method | Acc. (%) | Acc. gain | Time (s/epoch) | Mem (GiB) |
|---|---|---|---|---|---|
| Plain-20 | Vanilla | 90.18 ± 0.27 | – | 2.84 | 0.27 |
| | SGMS, $\beta = 0.9$ | 90.41 ± 0.27 | +0.23 | 2.85 (×1.00) | 0.27 (×1.00) |
| | SGMS, $\beta = 1$ (GC) | 90.44 ± 0.32 | +0.26 | 2.84 (×1.00) | 0.27 (×1.00) |
| | AdaBK | 91.39 ± 0.29 | +1.21 | 3.05 (×1.07) | 0.71 (×2.49) |
| ResNet-20 | Vanilla | 91.41 ± 0.19 | – | 3.00 | 0.29 |
| | SGMS, $\beta = 0.9$ | 91.65 ± 0.23 | +0.24 | 3.03 (×1.01) | 0.29 (×1.00) |
| | SGMS, $\beta = 1$ (GC) | 91.42 ± 0.16 | +0.01 | 3.02 (×1.01) | 0.29 (×1.00) |
| | AdaBK | 91.99 ± 0.12 | +0.58 | 3.22 (×1.07) | 0.71 (×2.36) |
| ResNet-32 | Vanilla | 92.04 ± 0.16 | – | 4.42 | 0.35 |
| | SGMS, $\beta = 0.9$ | 92.19 ± 0.19 | +0.15 | 4.45 (×1.01) | 0.35 (×1.00) |
| | SGMS, $\beta = 1$ (GC) | 92.14 ± 0.21 | +0.10 | 4.43 (×1.00) | 0.35 (×1.00) |
| | AdaBK | 92.83 ± 0.16 | +0.79 | 4.79 (×1.08) | 0.83 (×2.32) |
| ResNet-44 | Vanilla | 91.99 ± 0.25 | – | 5.90 | 0.44 |
| | SGMS, $\beta = 0.9$ | 92.29 ± 0.32 | +0.30 | 5.97 (×1.01) | 0.44 (×1.00) |
| | SGMS, $\beta = 1$ (GC) | 92.24 ± 0.35 | +0.25 | 5.91 (×1.00) | 0.44 (×1.00) |
| | AdaBK | 92.95 ± 0.28 | +0.96 | 6.33 (×1.07) | 0.95 (×2.11) |
| ResNet-56 | Vanilla | 92.16 ± 0.33 | – | 7.29 | 0.55 |
| | SGMS, $\beta = 0.9$ | 92.40 ± 0.22 | +0.24 | 7.35 (×1.01) | 0.55 (×1.00) |
| | SGMS, $\beta = 1$ (GC) | 92.19 ± 0.10 | +0.03 | 7.28 (×1.00) | 0.55 (×1.00) |
| | AdaBK | 93.02 ± 0.22 | +0.86 | 7.86 (×1.08) | 1.05 (×1.93) |

Table 8: Effect of activation functions on CIFAR-100. We report Top-1 accuracy (%) for ResNet-20/56 with Vanilla and SGMS. SGMS assumes nonnegative activations; improvements disappear when the assumption is violated (e.g., Tanh).

| Model | Method | ReLU | SiLU | GeLU | Tanh |
|---|---|---|---|---|---|
| ResNet-20 | Vanilla | 72.5 | 73.0 | 72.8 | 71.9 |
| | SGMS | 74.2 | 73.9 | 73.5 | 71.8 |
| ResNet-56 | Vanilla | 78.9 | 79.1 | 79.0 | 77.8 |
| | SGMS | 81.2 | 80.3 | 80.0 | 77.7 |

$$\Delta W = \begin{cases} (I - \mathbf{e}\mathbf{e}^{\top}) \Delta W, & \textbf{GC } (\beta = 1), \\ (I - \beta\,\mathbf{e}\mathbf{e}^{\top}) \Delta W, & \textbf{SGMS } (0 \leq \beta < 1), \\ \Delta W' \cdot \dfrac{\|\Delta W\|_F}{\|\Delta W'\|_F}, & \text{SGMS with optional rescaling.} \end{cases}$$

*Reporting policy.* - In all tables, "**SGMS,** $\beta = 1$ **(GC)**" denotes pure GC (i.e., no rescaling). - For **SGD**, we apply the optional rescaling step only when $\beta < 1$. - For **Adam/AdamW**, we omit rescaling entirely.

# E    RELATION TO ADABK

Although our initial motivation came from LoRA—controlling the effective update subspace—the method we arrived at turned out to be more closely connected to AdaBK Yong et al. (2023). In fact,

Table 9: Sensitivity to the scaling factor $\beta$ on CIFAR-100. We report Top-1 **final** accuracy (%) averaged over 5 runs (mean ± std). Best result is in **bold**, second best is underlined.

| $\beta$ | ResNet-20 (Acc.) | ResNet-56 (Acc.) |
|---|---|---|
| 0.0 (vanilla) | 66.98 ± 0.22 | 69.43 ± 0.76 |
| 0.1 | 67.67 ± 0.46 | 69.90 ± 0.57 |
| 0.2 | 67.43 ± 0.24 | 69.62 ± 0.46 |
| 0.3 | 67.44 ± 0.39 | 70.03 ± 0.46 |
| 0.4 | 67.44 ± 0.35 | 69.81 ± 0.84 |
| 0.5 | 67.17 ± 0.46 | 70.10 ± 0.75 |
| 0.6 | 67.57 ± 0.32 | 69.77 ± 0.55 |
| 0.7 | 67.52 ± 0.33 | 70.26 ± 0.45 |
| 0.8 | 67.81 ± 0.21 | **70.65 ± 0.58** |
| 0.9 | **67.93 ± 0.34** | 70.57 ± 0.40 |
| 1.0 | 67.78 ± 0.44 | 70.48 ± 0.66 |

Table 10: Effect of Frobenius-norm rescaling on CIFAR-100 (SGD): Top-1 *final* accuracy (%) over 5 runs (mean ± std). Under SGD we apply Frobenius-norm rescaling; under Adam/AdamW no rescaling is used.

| Model | w/o rescale | w/ rescale |
|---|---|---|
| ResNet-20 | 67.51 ± 0.37 | 67.93 ± 0.34 |
| ResNet-56 | 70.86 ± 0.58 | 70.57 ± 0.40 |

SGMS can be interpreted as a lightweight approximation to AdaBK. Both methods aim to mitigate the confinement of gradients to a low-rank subspace that arises from the covariance structure of the feature activations $X$ and the backpropagated signals $\Delta Y$.

AdaBK explicitly applies decorrelation to both sides by using the **inverse square roots** of the corresponding covariance matrices:

$$C_X = \tfrac{1}{n}X^\top X, \quad C_{\Delta Y} = \tfrac{1}{n}(\Delta Y)^\top(\Delta Y), \quad \Delta W_{\text{AdaBK}} = C_X^{-\frac{1}{2}}\,\Delta W\, C_{\Delta Y}^{-\frac{1}{2}}, \quad (16)$$

which balances the contributions across all singular directions of $X$ and $\Delta Y$. This full preconditioning is effective but incurs non-trivial computational cost due to the SVD or iterative operations needed to obtain $C_X^{-\frac{1}{2}}$ and $C_{\Delta Y}^{-\frac{1}{2}}$ during training.

By contrast, SGMS performs a single, computationally lightweight adjustment—subtracting the column mean from each column of $\Delta W$—optionally followed by a norm-preserving rescale (Eq. 15). This selectively suppresses the dominant uniform direction, which under nonnegative activations (e.g., ReLU) serves as a practical proxy for the leading singular vector. Its limitations are that the effect is generally weaker than AdaBK's full preconditioning and that it relies on $X$ being the output of a nonnegative activation function.

## F  APPLYING SGMS TO CONVOLUTIONAL AND ATTENTION LAYERS

**Convolutional layers.** For convolutional layers, SGMS is applied to the weight gradient in the same way as for fully-connected layers. Formally, the gradient of the kernel has shape

$$\Delta W \in \mathbb{R}^{(c_{\text{in}}h_k w_k)\times c_{\text{out}}},$$

where $c_{\text{in}}$ and $c_{\text{out}}$ are the input and output channel counts, and $h_k \times w_k$ is the kernel size. If the convolution is expressed in "matrix multiplication" form via the standard `im2col` transformation, then the input is unfolded into a matrix of size $(nh_f w_f) \times (c_{\text{in}}h_k w_k)$, where $h_f \times w_f$ denotes the spatial resolution of the output feature map for a batch of size $n$. The convolution becomes an ordinary matrix multiplication with $\Delta W$ as above. Thus SGMS can be applied to convolutional layers exactly as in the fully-connected case: subtracting the column-wise mean of $\Delta W$ and rescaling if

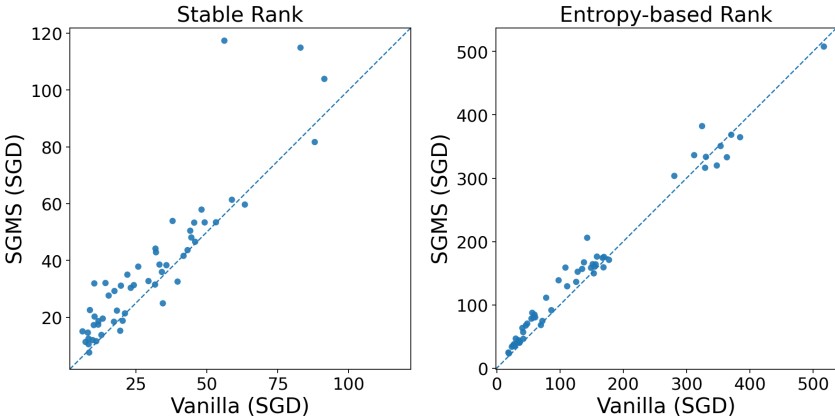

Figure 6: Layer-wise comparison of spectral dispersion in ResNet-50 trained on ImageNet. Each point corresponds to a single linear transformation (fully connected or convolutional layer unfolded into a matrix), for which we compute the spectral ranks of its weight matrix. **(Left)** stable rank, **(Right)** entropy-based rank. In both metrics, most points lie above the diagonal ($y$=$x$), indicating that SGMS increases the rank of learned weight matrices and distributes spectral energy across more singular directions.

needed. This equivalence means no special treatment is required for convolutional layers beyond reshaping.

**Attention layers.** The core computations in self-attention also reduce to matrix multiplications, including the linear projections ($Q = XW_Q$, $K = XW_K$, $V = XW_V$) and the pairwise product $QK^\top$. Therefore, SGMS can be applied to the projection matrices ($W_Q$, $W_K$, $W_V$) in exactly the same way as in fully-connected layers. In particular, when the input to these projections follows a nonnegative activation function, the same theoretical justification as in Section 3.4 applies, and SGMS can be used without modification.

# G  THEORETICAL OVERHEAD (MULTIPLICATION-ONLY ACCOUNTING)

We count only the number of *multiplications*, which dominate runtime on modern accelerators. For each layer, the main operation (a matrix multiplication in fully-connected layers or a convolution in CNN layers) appears three times per iteration: once in the forward pass and twice in the backward pass (once to form the weight gradient and once to propagate the gradient to the inputs). By contrast, *SGMS is executed once* after the backward pass has produced the weight gradient $\Delta W$.

**Fully-connected layer.** Let the mini-batch size be $n$, the **input** dimension be $k$, and the output dimension be $d$. The forward pass computes $Y = XW$ and costs $nkd$ multiplications. The backward pass performs two large matrix multiplications, $\Delta W = X^\top \Delta Y$ and $\Delta X = \Delta Y W^\top$, costing $2nkd$ multiplications. Hence the baseline (forward + backward) totals

$$\text{Mults}_{\text{base, fc}} = 3nkd.$$

SGMS acts on $\Delta W \in \mathbb{R}^{k \times d}$ by subtracting the column-wise mean and (optionally) rescaling. Counting only multiplications: (i) column-wise mean scaling uses $d$ multiplications (a factor $1/k$ per column); (ii) $\|\Delta W\|_F$ uses $kd$ multiplications (squares); (iii) $\|\tilde{\Delta W}\|_F$ uses $kd$ multiplications; and (iv) rescaling $\tilde{\Delta W}$ uses $kd$ multiplications. Thus

$$\text{Mults}_{\text{SGMS, fc}} = 3kd + d.$$

The relative overhead is

$$\frac{\text{overhead}}{\text{baseline}} = \frac{3kd + d}{3nkd} = \frac{1}{n} + \frac{1}{3nk} \approx \frac{1}{n}.$$

For $n$=256, this is about $0.39\%$ (the $\frac{1}{3nk}$ term is negligible for moderate $k$). (If rescaling is omitted, the SGMS cost drops to $\approx d$, making the ratio even smaller.)

**Convolutional layer.** Let the input have $c_{\text{in}}$ channels, the output have $c_{\text{out}}$ channels, the output feature map size be $h_f \times w_f$, and the kernel size be $h_k \times w_k$. The forward convolution costs

$$n\, h_f w_f\, c_{\text{out}}\, c_{\text{in}} h_k w_k$$

multiplications, and the two backward convolutions (for $\Delta W$ and $\Delta X$) have the same order, so the baseline totals

$$\text{Mults}_{\text{base, conv}} \;=\; 3\, n\, h_f w_f\, c_{\text{in}} h_k w_k\, c_{\text{out}}.$$

SGMS again acts on $\Delta W$ of shape $c_{\text{in}} h_k w_k \times c_{\text{out}}$ and costs

$$\text{Mults}_{\text{SGMS, conv}} \;=\; 3\, c_{\text{in}} h_k w_k\, c_{\text{out}} \;+\; c_{\text{in}} h_k w_k$$

multiplications (column-wise mean scaling, two Frobenius norms, and rescaling). Therefore,

$$\frac{\text{overhead}}{\text{baseline}} \;=\; \frac{3\, c_{\text{in}} h_k w_k\, c_{\text{out}} + c_{\text{in}} h_k w_k}{3\, n\, h_f w_f\, c_{\text{in}} h_k w_k\, c_{\text{out}}} \;=\; \frac{1}{n h_f w_f} \;+\; \frac{1}{3 n h_f w_f\, c_{\text{out}}}.$$

Because the factor $h_f w_f$ (the number of output spatial locations) appears in the denominator, the ratio is much smaller than in the fully-connected case. For example, with $n=256$ and $h_f w_f = 14 \times 14 = 196$, the leading term is $1/(256 \cdot 196) \approx 2.0 \times 10^{-5}$ ($\sim$0.002%).

For ResNet-20, the theoretical overhead of SGMS relative to the total multiply count of forward and backward passes is approximately 0.0026% for a standard batch size of 256.

**Remark on optimizers.** For SGD, the optional rescaling step is theoretically aligned with the purpose of SGMS (Section C.4) and is therefore included in the above cost estimates. In contrast, when using adaptive optimizers such as Adam, the effect of rescaling is naturally absorbed into the learning rate adjustment. In this case, SGMS only requires computing and subtracting the column-wise mean of $\Delta W$, whose cost is negligible compared to the baseline matrix multiplications.

## H  JUSTIFICATION OF UNIFORM DIRECTION APPROXIMATION

Let $X \in \mathbb{R}^{n \times k}$ consist of i.i.d. nonnegative rows $\boldsymbol{x}_i$ satisfying $\mathbb{E}[\boldsymbol{x}_i] = m\mathbf{1}$ for some $m > 0$, where $\mathbf{1} := (1, \ldots, 1)^\top \in \mathbb{R}^k$. Let $x_{ij}$ denote the $j$-th entry of $\boldsymbol{x}_i$, and define

$$m = \mathbb{E}[x_{ij}], \qquad s^2 := \mathbb{E}[x_{ij}^2].$$

Assuming that the centered coordinates $(x_{ij} - m)$ are approximately uncorrelated with common variance $s^2 - m^2$, we have

$$\mathbb{E}[\boldsymbol{x}_i \boldsymbol{x}_i^\top] \approx (s^2 - m^2) I_k + m^2\, \mathbf{1}\mathbf{1}^\top,$$

where $I_k$ represents an identity matrix.

Hence the expected feature Gram matrix

$$A := \mathbb{E}[X^\top X] = a I_k + b\, \mathbf{1}\mathbf{1}^\top, \qquad a = n(s^2 - m^2), \quad b = n m^2.$$

Let $\boldsymbol{e} := \frac{1}{\sqrt{k}}\mathbf{1}$. Then

$$A\boldsymbol{e} = a\boldsymbol{e} + b\, \mathbf{1}\mathbf{1}^\top \boldsymbol{e} = a\boldsymbol{e} + bk\, \boldsymbol{e} = (a + bk)\boldsymbol{e},$$

so $\boldsymbol{e}$ is an eigenvector of $A$ with eigenvalue $a + bk$. Since $A = a I_k + b\, \mathbf{1}\mathbf{1}^\top$ is a rank-one perturbation of $a I_k$, its remaining $k - 1$ eigenvalues are all equal to $a$. Thus $a + bk$ is strictly larger than all other eigenvalues, and $\boldsymbol{e}$ is the unique leading eigenvector of $A$. Its eigenvalues satisfy

$$\lambda_1(A) = a + bk, \qquad \lambda_2(A) = a, \qquad \lambda_1(A) - \lambda_2(A) = n m^2 k.$$

Now let $\hat{X}$ be the empirical feature matrix obtained from a mini-batch, and define

$$B := \hat{X}^\top \hat{X}.$$

We write

$$B = A + R,$$

where $R$ collects the sampling deviation.

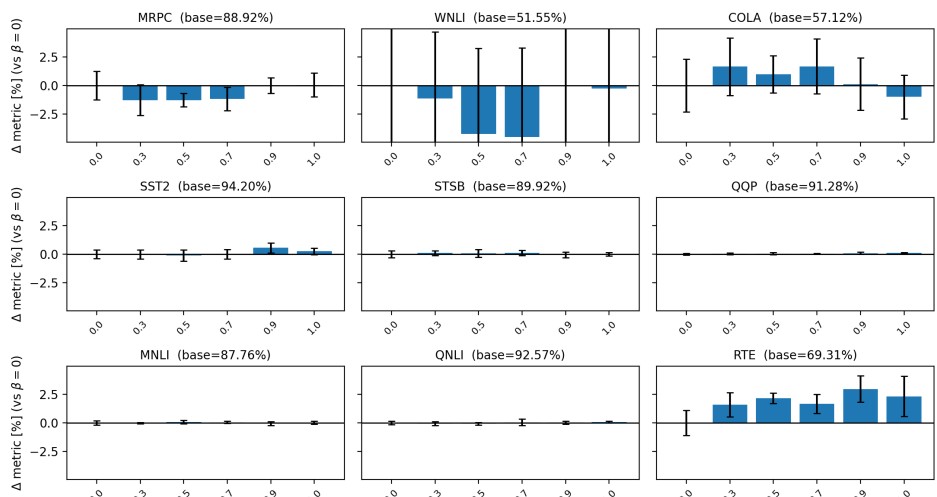

Figure 7: **GLUE fine-tuning results for RoBERTa-base.** Each panel shows the change in validation performance (accuracy/F1/Spearman), relative to the vanilla baseline ($\beta = 0$), for a single GLUE task.

**Davis–Kahan sin$\Theta$ theorem.** The standard form of the Davis–Kahan bound gives

$$\sin \angle(\boldsymbol{v}_1, \boldsymbol{e}) \ \leq \ \frac{\|R\|_2}{\lambda_1(A) - \lambda_2(A)} = \frac{\|R\|_2}{nm^2k},$$

where $\| \cdot \|_2$ represents the spectral norm, and $\sin \angle(\boldsymbol{v}_1, \boldsymbol{e})$ is the sine of the angle between the two vectors.

Since the eigenvalue gap of $A$ is $\lambda_1(A) - \lambda_2(A) = nm^2k$, which is typically large in practical training settings, the denominator of the Davis–Kahan bound becomes substantial. Therefore, even if the sampling deviation $\|R\|_2$ is moderately large, the top empirical feature direction $v_1$ is still guaranteed to remain close to the uniform direction $e$. This theoretical behavior is consistent with our empirical observations that the leading feature direction of $X^\top X$ is nearly uniform.

# I  EMPIRICAL VERIFICATION OF UNIFORM-DIRECTION ALIGNMENT IN GELU ACTIVATIONS

To verify that SGMS remains applicable under GeLU, we examined the hidden representations of a pretrained RoBERTa model. For each linear layer immediately after a GeLU activation, we formed the empirical feature matrix $X$ and computed its leading right singular vector $\boldsymbol{v}_1$. The alignment with the uniform direction $\boldsymbol{e}$ was substantial; for example, in the first encoder block we observed $|\boldsymbol{v}_1^\top \boldsymbol{e}| = 0.8346$. This confirms that GeLU activations in practice induce a strong dominant direction closely aligned with $\boldsymbol{e}$, consistent with the assumptions underlying SGMS.

## I.1  RESULTS OF ROBERTA FINETUNED TO GLUE DATASET

To examine whether SGMS transfers beyond vision models, we conducted additional experiments on GLUE (Wang et al., 2019) using RoBERTa (Liu et al., 2019) and DistilBERT (Sanh et al., 2019). For each model, we fine-tuned on all nine GLUE tasks with $\beta \in \{0, 0.3, 0.5, 0.7, 0.9, 1.0\}$, using five random seeds following the standard training recipe in the HuggingFace implementation. We report the average validation accuracy over seeds, and visualize the change relative to the vanilla baseline ($\beta = 0$) in Figure 7 and 8 using task-wise bar plots.

Across eight tasks—excluding WNLI, for which learned models are known to perform below most-frequent-class guessing—we observe that SGMS does not significantly degrade performance. In several tasks (e.g., CoLA and RTE with RoBERTa), SGMS ($\beta = 0.3$–$0.7$ for CoLA and $\beta = 0.3$–$0.9$ for RTE) even yield slight improvements, although the effect size seems within the variance of seed

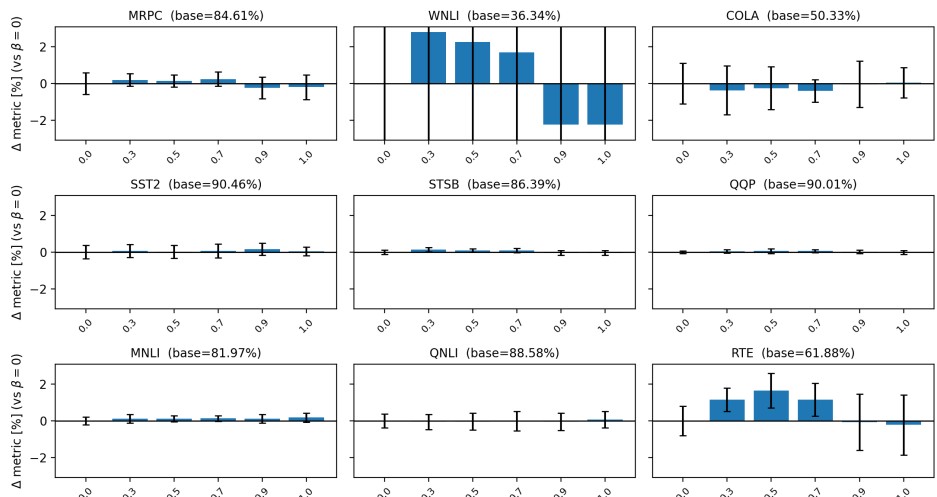

Figure 8: **GLUE fine-tuning results for DistilBERT.** Each panel shows the change in validation performance, relative to the vanilla baseline ($\beta = 0$), for each GLUE task.

sampling in most cases. DistilBERT shows a similar pattern with particularly stable behavior for $\beta = 0.3$–$0.7$. Overall, SGMS does not degrade performance in pretrained NLP models. This small effect is unsurprising: strong pretrained representations already exhibit well-behaved feature geometry, leaving little room for SGMS to make a noticeable impact.

## J   LLM USAGE

Large language models (LLMs) were used solely as a writing assistant to polish the presentation of this paper. They were not involved in generating research ideas, designing experiments, analyzing results, or drawing conclusions. All scientific contributions, including problem formulation, method design, experiments, and interpretations, are solely the work of the author. The author takes full responsibility for the contents of the paper.

