# OpenReview forum: "Scaled Gradient Mean Subtraction: A Lightweight Method for Amplifying Underutilized Gradient Directions"
_ICLR.cc/2026/Conference — Submitted to ICLR 2026_

### Official Review · Reviewer_d3aU · 2025-10-31

**Soundness:** 2
**Presentation:** 2
**Contribution:** 2
**Rating:** 2
**Confidence:** 4

**Summary:**

This paper introduces a method termed Scaled Gradient Mean Subtraction (SGMS), which aims to enhance neural network training by mitigating the influence of dominant gradient directions. The core idea involves scaling the mean of the gradient before its subtraction. The authors provide an observation that in layers with non-negative activations (e.g., ReLU), this dominant direction frequently aligns with the uniform vector, thus framing SGMS as an efficient approximation for suppressing the dominant singular value. A significant advantage of this approach is its computational efficiency, as it avoids the expensive matrix decompositions required by preconditioners like AdaBK. The empirical evaluation covers CNNs and Vision Transformers on CIFAR and ImageNet benchmarks. The reported results consistently demonstrate accuracy improvements over standard training, supporting the efficacy of the proposed method.

**Strengths:**

The motivation is clear, the authors take an insight into subspace concentration of gradient updates, and  analyze ‘why weight updates naturally concentrate in a low-rank subspace’ and provide evidences of how SGMS approximate the decomposition process, even though I have concerns on its novelty.  The writing is clear and the paper is easy to follow.

**Weaknesses:**

1. **The novelty of SGMS is limited and a bunch of related work are missed**. Firstly, it is highly consistent with Gradient Centering (GC).  Without demonstrating a notable improvement over GC, the introduction of an extra hyper-parameter ($\beta$) may even reduce its practical utility. Secondly,  the idea of centering can theoretically improve the conditioning (i.e., suppressing the dominant singular value) can at least track back to the paper [1] in 1998, and following work [2,3]. This paper miss the related work along this track.

   Besides, the field's focus has shifted towards covariance-based pre-conditioners because they decouple gradient directions, whereas centering operations primarily make the gradients well-distributed and only approximately utilizes gradients better. In this way, the formulation of this method is more like the recently work in gradient orthogonalization[4], or projected based gradient updates for  LLM training, e..g, the Galore[5]. I think this paper should provide a detailed comparison to these methods.



2. **Concerns on the experiments :**

   (1) The empirical validation is limited, covering only image classification tasks (CIFAR, ImageNet-1k) with a limited set of models (ResNet, DeiT-Small). Broader tasks and architectures are needed to demonstrate generality.

   (2) The authors claim GC is a "special case" of SGMS, yet the ImageNet-1k experiments lack a critical comparison between Vanilla, GC, and SGMS. This makes it difficult to assess the actual performance gap and the value of the proposed scaling factor.

   (3) The authors suggest that $\beta$ should be chosen statistically based on singular values (Line 286-290). However, in practice, it is treated as a fixed hyperparameter without a learnable or strategic selection process. Furthermore, the evidence for a "good" $\beta$ is insufficient and is only provided on small-scale experiments (ResNet on CIFAR10), not on the more challenging settings like ImageNet-1k or DeiT.

3. **Clarity of Assumption:** The authors' central claim that GC/SGMS balances gradient distribution seems to rely on an unstated assumption: "The greater the activation value after a nonnegative activation function, the greater the gradient, and the more the distribution." Making this assumption explicit is crucial for the reader's understanding.

4. **Justification of Claims:** The observation in Line 143-144 that SGMS "amplifies underutilized directions" is interesting but not fully substantiated. While it may be ignored by LoRA, it does not prove that these directions are useful rather than being noise. Further analysis is required to validate their utility.

5. **Typo:** Table 9 is missing the conventional bold and underline formatting, which should be corrected.



   **Ref:**

   [1] Accelerated Gradient Descent by Factor-Centering Decomposition, 1998

   [2] Deep Boltzmann Machines and the Centering Trick,  2012

   [3] Mean-normalized stochastic gradient for large-scale deep learning. ICASSP, 2014

   [4] SWAN: SGD with Normalization and Whitening Enables Stateless LLM Training, ICML, 2025.

   [5] GaLore: Memory-Efficient LLM Training by Gradient Low-Rank Projection, ICML, 2024

**Questions:**

See Weaknesses.

**Details Of Ethics Concerns:**

NA.

---

> ### Author Response · Authors · 2025-11-22
> **Responses to comments from d3aU**
>
> We appreciate the constructive review.
>
> # Answer to Q1.
>
> Because this question concerns the conceptual positioning of SGMS relative to
> prior methods, we provide a detailed clarification below. We hope this helps
> clarify the novelty and intended scope of our contribution.
>
> We acknowledge that the two components underlying SGMS have each been discussed
> independently in earlier work:
> (a) LeCun et al. [1] noted that suppressing the dominant singular direction can
>     accelerate optimization when second-order information is available, and
> (b) gradient centering (GC) has been reported to improve training stability.
>
> However, these ideas remained separate. Our contribution is to show that, in
> modern networks with ReLU/GeLU activations, the column-wise mean of $\Delta W$ closely
> approximates the dominant right singular direction $v_1$ of the feature matrix $X$.
> This activation-induced structure provides a geometric explanation for why mean
> subtraction affects optimization and reveals that GC implicitly removes this
> dominant direction entirely. This link does not appear in prior work and is what
> enables SGMS: recognizing this structure naturally leads to the idea of partial
> suppression, retaining useful components while mitigating the activation-induced
> rank-1 bias.
>
> Regarding [2], although this work also uses “centering,” it modifies the forward
> activations and removes the mean entirely, without considering the
> feature-dimension structure of the weight gradient. SGMS, by contrast, leaves the
> forward pass unchanged and acts only on the gradient, targeting the
> activation-induced dominant direction.
>
> Regarding [3], the per-coordinate minibatch centering considered there operates
> along the sample axis and does not examine correlations across neurons or
> features. It therefore does not affect the singular-value geometry that SGMS
> targets.
>
> Regarding SWAN [4], this method performs full-matrix normalization and whitening
> on gradient blocks by estimating covariance matrices. Its goal is to stabilize
> stateless SGD in LLMs. SGMS does not whiten gradients or estimate second-order
> statistics; it removes only a single fixed direction with one mean subtraction.
> The mechanism, cost, and intended use case are fundamentally different.
>
> Regarding GaLore [5], this method projects gradient matrices onto a learned
> low-rank subspace to reduce optimizer-state memory in LLM training. SGMS does not
> learn subspaces, maintain gradient history, or perform low-rank projections.
> GaLore and SGMS therefore address different goals and operate in different
> regimes.
>
> For methods aiming to improve conditioning through spectral effects, the closest
> and most meaningful comparison is the family of covariance-based preconditioners
> such as AdaBK, which directly manipulate the gradient spectrum. SGMS provides a
> lightweight alternative to this family, capturing the leading-direction
> suppression effect without forming or inverting covariance matrices. For this
> reason, AdaBK is used as the representative baseline.
>
> However, we have added the cited the suggested references to the Related Work section.

---

> ### Author Response · Authors · 2025-11-22
>
> # Answer to Q2.
>
> ## (1) Regarding the experimental scope:
> Our evaluation follows the standard protocol used in prior work on gradient
> manipulation and conditioning methods such as Gradient Centralization, AdaBK,
> Shampoo, etc.—namely CIFAR and ImageNet-1k on both CNNs (ResNet) and
> Transformers (DeiT-Small). These are the de-facto benchmarks for assessing the
> generality of optimization techniques across architectures.
>
> Importantly, SGMS is a layer-local gradient transformation that does not rely on
> any task-specific structure. Its effectiveness derives from a structural property
> of modern architectures—the asymmetric, nonnegative post-activations induced by
> ReLU-like nonlinearities—which is widespread across vision and transformer
> models. Thus the method is inherently task-agnostic.
>
> We also added RoBERTa and DistilBERT finetuning experiments on GLUE. As expected,
> the effect is smaller in this regime because pretrained models already exhibit
> well-shaped representations, leaving limited room for gradient-geometry methods
> to influence the optimization dynamics.
>
> ## (2) Regarding the comparison with GC:
> We have added full ImageNet-1k results including the GC baseline. SGMS consistently
> outperforms both vanilla training and GC, indicating that partial suppression via
> $\beta$ provides a clear benefit over removing the entire mean component.
>
> ## (3) Regarding the choice of $\beta$:
> Lines 283-288 in the revised paper describe that, in principle, an *ideal* $\beta$ could be chosen by
> maximizing a criterion such as spectral entropy of $X$. However,
> estimating singular values or covariance matrices every iteration would incur
> overhead comparable to full preconditioners (e.g., AdaBK), defeating
> the lightweight goal of SGMS. For this reason, we treat $\beta$ as a simple scalar
> hyperparameter.
>
> The optimal $\beta$ may vary across different architectures, datasets, and so on.
> However, in our experiments, the value selected from CIFAR ($\beta=0.9$) transfers stably to ImageNet. In all ImageNet-1k
> experiments, $\beta=0.9$ consistently improves over both vanilla and GC without any
> tuning. This indicates that SGMS is not overly sensitive to $\beta$ in large-scale training.
>
> # Answer to Q3.
>
> Thank you for raising this point. To clarify, SGMS does not rely on the
> assumption that “larger post-activation values lead to larger gradients.”
> This relationship may hold in some architectures, but it is not the basis of
> our analysis.
>
> The structural assumption we rely on is the one stated in the paper:
> after nonnegative (or positively biased) activations such as ReLU or GeLU,
> the feature matrix $X$ tends to lie in the positive orthant, which induces a
> dominant rank-1 component whose right singular vector aligns with the uniform
> direction $e$. This phenomenon concerns the *geometry* of $X$ and $\Delta W = X^\top \Delta Y$,
> not the absolute magnitudes of individual gradient entries.
>
> # Answer to Q4.
>
> We appreciate the reviewer’s point. Indeed, underutilized directions may
> contain noise, and our claim is not that every such direction is inherently
> useful.
>
> The intended meaning is the one widely used in the literature on
> covariance-based preconditioners (e.g., Shampoo, K-FAC, AdaBK): when the
> gradient spectrum is highly unbalanced, suppressing the overly dominant
> direction restores a more diverse set of descent directions and improves the
> conditioning of the optimization problem. These works consistently show that
> flattening the singular-value spectrum improves optimization efficiency and
> generalization, even though individual directions may contain noise.
>
> SGMS can be viewed as a lightweight counterpart to these methods. Instead of
> computing or inverting covariance matrices, SGMS reduces the dominance of the
> largest singular component using one mean subtraction. This naturally increases
> the relative contribution of the remaining directions in the established sense
> of improving gradient diversity, without asserting that each direction is
> signal-bearing.
>
> The empirical gains we observe on CIFAR and ImageNet indicate that this
> rebalancing is beneficial in practice.
>
> # Answer to Q5.
> We formatted it in that way. Thank you for pointing this out.

---

### Official Review · Reviewer_G3oa · 2025-11-01

**Soundness:** 2
**Presentation:** 3
**Contribution:** 2
**Rating:** 4
**Confidence:** 4

**Summary:**

This paper proposes Scaled Gradient Mean Subtraction (SGMS), a simple method to improve neural network training by rebalancing gradient directions to use underutilized directions. The authors observe that mini-batch gradients often lie in a low-rank subspace dominated by a few directions, limiting optimization efficiency. SGMS subtracts a scaled mean from layer gradients to partially suppress the dominant gradient direction (unlike full Gradient Centralization (GC)), thereby amplifying underutilized directions and potentially improving generalization. Experiments on Cifar and ImageNet show consistent small accuracy gain for SGMS.

**Strengths:**

1.	The paper structures an argument connecting gradient subspace degeneracy to underutilization and introduces SGMS as a low-cost alternative to heavy preconditioning methods.
2.	Results on multiple datasets (CIFAR, ImageNet) and architectures (ResNets, DeiT), show that SGMS improves performance.

**Weaknesses:**

1.	The authors present SGMS as a generalization of GC, using partial rather than full suppression of dominant gradient directions. However, this modification is conceptually minor and lacks deeper analytical justification for why or when the partial suppression (β < 1) works better, the method is indeed a parametric relaxation of GC with some interesting thought.
2.	The paper lacks a theoretical guarantee that SGMS stabilizes training. Some parts of the theoretical discussion is based on intuition and observation, for example, the authors approximate the dominant singular vector with the uniform vector e, empirically true for BatchNorm–ReLU networks but may not generally apply to architectures lacking normalization or using zero-centered activations.
3.	The performance gains are limited (mostly between 0.2-0.8 %), which is expected given that β=0.9 is very close to β=1 in GC. There is now evidence or theoretical analysis explaining why higher β values perform better. On the other hand, given that AdaBK achieves higher accuracy, SGMS’s contribution appear limited.
4.	Lack of long training, or larger transformer architectures (e.g., ViT-B) undermines the claimed robustness and generality. Moreover, all experiments are conducted from scratch, with no investigation of cross-task or transfer-learning scenarios such as fine-tuning pretrained backbones, which limits understanding of SGMS behavior in practical pipelines.
5.	The ablation study lacks analysis of optimizer dynamics (e.g., with momentum, weight decay, or learning rate scaling), as well as evaluations under large-batch training.

**Questions:**

1. Can you theoretically justify how mean subtraction (Eq. 12) affects convergence or stability?

2. What is the rationale behind choosing β = 0.9, and could smaller β values be beneficial in cases with more correlated gradients? Is there any theoretical guidance for selecting β instead of purely empirical tuning?

3. Have you verified whether the assumption v_1≈e holds for networks without normalization (e.g., LSTM-based architectures)?

4. Can you examine SGMS robustness across optimizer dynamics?

5. Does SGMS help in under-parameterized or small-data regimes, where gradient diversity is naturally lower?

---

> ### Author Response · Authors · 2025-11-22
> **Responses to comments from G3oa**
>
> We appreciate the constructive review.
>
> Before answering to the questions, we acknowledge that our submission may not have made this point sufficiently clear, we here clarify the source of the $v_{1} \approx e$ effect.
>
> The observed alignment $v_{1} \approx e$ does *not* come from BatchNorm or LayerNorm.
> It arises from the biased output distribution created by asymmetric activations such as ReLU or GeLU: these functions produce non-negative, one-sided feature vectors, which in turn yield an matrix whose dominant right singular vector aligns closely with the uniform direction. As a result, in the following linear layer, a large portion of the gradient energy is concentrated along this uniform direction.
>
> Because this effect is induced *solely* by the activation, SGMS is applicable to linear layers that directly follow asymmetric activations such as ReLU or GeLU.
> If a normalization layer intervenes between the activation and the linear layer, this bias no longer appears, and SGMS is not expected to be effective.
>
> # Answer to Q1.
>
> We would like to clarify the theoretical perspective behind Eq. (12).
> A well-known idea in optimization is that when gradient directions are more balanced, the optimization landscape becomes easier to navigate, which leads to faster and more stable convergence.
> This idea underlies a wide range of existing second-order or preconditioned optimizers, including Shampoo, K-FAC, and AdaBK (cited in the paper), all of which explicitly attenuate overly dominant directions while relatively strengthening underutilized ones.
>
> SGMS should be viewed as an extremely lightweight, rank-one approximation of this general singular-spectrum–flattening principle.
> As we also explained in our response to another reviewer, our empirical analysis shows that after ReLU-like activations, the gradient exhibits a strong rank-1 bias:
> the dominant right singular vector $v_{1}$ is very likely to be closely aligned with the uniform direction $e$ (please see Appendix I).
> This phenomenon arises from the one-sided, asymmetric nature of these activation functions and is consistently observed across architectures and datasets.
>
> Given this structure, the effect of SGMS becomes straightforward to interpret.
> Subtracting the column mean from the gradient matrix removes (approximately) the component along its dominant singular direction.
> Spectrally, this operation shrinks only the largest singular value while leaving the remaining singular values essentially unchanged, thereby flattening the singular-value spectrum at negligible computational cost.
>
> Thus, the convergence behavior of SGMS does not require a new convergence theory:
> it naturally aligns with the extensive body of work showing that flattening the singular-spectrum leads to faster and more stable optimization.
>
> # Answer to Q2.
>
> The value $\beta = 0.9$ was selected based on a fine-grained grid search on CIFAR-100,
> where we swept $\beta \in \{0.0, 0.1, \dots, 1.0\}$ using ResNet-style architectures (see Appendix C3).
> Values around $\beta \approx 0.9$ consistently gave the strongest improvements in this setting,
> so we simply carried this choice over to ImageNet \emph{without any additional tuning}.
> Using the CIFAR-selected value still yielded clear accuracy gains on DeiT-Small and other ImageNet models.
>
> From a theoretical perspective, SGMS can be viewed as encouraging a flatter singular-value spectrum of the layer input $X$.
> In principle, an “ideal’’ $\beta$ could be obtained by maximizing a measure of spectral entropy of $X$.
> However, estimating such quantities would require computing singular values or covariance matrices of $X$ at every iteration,
> introducing overhead comparable to second-order preconditioners such as Shampoo or AdaBK.
> To keep SGMS lightweight, we simply treat $\beta$ as a hyperparameter and select a fixed value.
>
> We have revised the paragraph around line 285 (paragraph beginning with "Ideally, the coefficient ...") to clarify how we choose $\beta$ and why we treat it as a lightweight hyperparameter.

---

> ### Author Response · Authors · 2025-11-22
>
> # Answer to Q3.
>
> The assumption $v_{1} \approx e$ does not rely on the presence of normalization
> layers.  Instead, it arises from the asymmetric shape of modern activations such
> as ReLU or GeLU, which induce a strong rank-1 bias in the post-activation feature
> matrix $X$.  When a linear layer directly follows such an activation, the rows
> of $X$ become positively aligned, and the dominant right singular vector
> naturally becomes close to the uniform direction $e$.  This mechanism holds
> regardless of whether normalization layers (BatchNorm, LayerNorm) are present, as
> long as they do not appear between the activation and the linear layer of
> interest.
>
> To clarify, the presence or absence of normalization layers does not create
> the $v_{1} \approx e$ structure.
> The alignment instead comes from the asymmetric shape of the activation
> (ReLU/GeLU) itself, and therefore the phenomenon can arise even in architectures
> that do not use normalization between the activation and the linear layer.
>
> In contrast, architectures using symmetric activations (e.g., Tanh) do not produce
> this one-sided bias, and the alignment need not hold.  Our method is therefore
> intended for modern ReLU/GeLU-based architectures rather than models
> with tanh activations.
>
> # Answer to Q4.
>
> We agree that examining SGMS under a wider range of optimizers and training
> regimes is valuable.  In the revised paper, our current experiments already include long training—most notably the standard 300-epoch ImageNet schedule for DeiT-Small,
> which is widely regarded as a long-training regime in modern vision benchmarks.
> Under this setting, SGMS consistently improves over both the vanilla baseline and
> GC, indicating that its effect does not vanish under extended optimization.
>
> Regarding robustness across optimizer dynamics, SGMS operates as a lightweight
> post-processing step applied to the gradient and therefore interacts only weakly
> with the specific optimizer update rule.  Empirically, we observe consistent gains
> across SGD-based training (ResNet/ImageNet, CIFAR), Adam-based training (CIFAR),
> and AdamW-based training (DeiT-Small). SGMS behaves similarly across all of them.
>
> A comprehensive evaluation over all optimizer families is beyond the scope of
> this work, but the evidence from these diverse settings already suggests that
> SGMS is stable and largely optimizer-agnostic.
>
> # Answer to Q5.
>
> Whether SGMS helps in a particular regime is determined not by the model size or
> the amount of data, but by the geometry of the feature matrix $X$.
> SGMS is effective when the gradients exhibit a clear dominant direction—i.e.,
> when $X$ shows a strong rank-1 bias induced by asymmetric activations such as
> ReLU or GeLU.
>
> In under-parameterized or small-data settings, the structure of $X$ can vary
> greatly depending on model capacity, data complexity, and noise.
> Such regimes may exhibit anything from nearly isotropic gradients to moderately
> structured ones, and therefore no universal prediction can be made about the
> impact of SGMS.  Conceptually, when a dominant direction does not reliably
> emerge, we do not necessarily expect SGMS to provide measurable improvements.
>
> We agree that exploring a broader range of regimes is valuable, but conducting a
> systematic survey over all combinations of model capacity, data size, and
> gradient structure remains an interesting direction for future work.

---

### Official Review · Reviewer_4W9V · 2025-11-01

**Soundness:** 3
**Presentation:** 3
**Contribution:** 3
**Rating:** 6
**Confidence:** 4

**Summary:**

The paper introduces Scaled Gradient Mean Subtraction (SGMS),
a simple gradient regularization method that subtracts a fraction
$\beta$ of each column’s mean from layer gradients.

It generalizes Gradient Centralization ($\beta=1$) and approximates
a lightweight form of spectral flattening. SGMS improves optimization stability
and test accuracy on Resnets (CIFAR, ImageNet) with minimal computational cost,
showing modest but consistent gains. The experimental section also includes results on transformer models,
where improvements are smaller but generally positive.

**Strengths:**

- $S_1$: SGMS is a one-line modification applicable to any optimizer and architecture, requiring negligible additional computation or memory. Appendix F details how to extend the intuitive SGMS to convolutions and attention layers and Appendix G details the theoretical overhead of the method. They both complement well the main text.

- $S_2$: The paper provides a clear theoretical motivation based on a linear-algebraic view of gradient formation ($\Delta W = X^\top \Delta Y$), connecting SGMS to both Gradient Centralization and spectral preconditioning. I would have liked a maybe more detailed paragraph on preconditionners in the related works section as structured second-order and preconditioning methods such as
K-FAC, Shampoo, or SOAP, are closely related to this line of work (together with AdaBK). However, given the page limitation, this is understandable.

- $S_3$: Empirical results on CIFAR and ImageNet ResNets show modest but consistent accuracy improvements with minimal cost. SGMS outperforms Gradient Centralization and approaches AdaBK performance at a fraction of its compute and memory.

- $S_4$: The method is empirically validated to be orthogonal to optimizer choice, showing compatibility and consistent benefits with SGD, Adam, and AdamW.

- $S_5$: The paper is transparent and thorough, reporting all hyperparameters, efficiency metrics, and activation-type dependencies, with well-designed ablations and diagnostics. Furthermore, it is enjoyable to read as the methodology section progressively walks us through the authors' intuition.

- $S_6$: Overall, this paper offers a clean conceptual unification between mean-subtraction methods and curvature-based preconditioners through the tunable scaling factor $\beta$, bridging Gradient Centralization and full whitening.

**Weaknesses:**

- $W_1$: The theoretical justification relies on the heuristic approximation $v_1 \approx e$ (alignment between the leading singular vector of $X$ and the uniform direction). This assumption is plausible but not formally justified; a perturbation bound or regularity assumption on $X^\top X$ would strengthen the argument.

- $W_2$: Transformer evaluation is limited to a shortened 100-epoch DeiT-Small schedule, which does not fully test the method under standard long-training regimes. Claims of generality beyond ResNets therefore remain preliminary. Furthermore, I would really like to see if this simple (yet efficient!) modification of the gradient transfers to other tasks than vision.
Experiments on NLP as (i) pretraining a small GPT-2 like transformer on OpenWebText (or similar dataset) and/or (ii) fine-tuning of a RoBERTa-like transformer on canonical tasks such as GLUE would be a strong plus to the paper and I would be happy to raise my score if the authors presented them before the end of the discussion phase.

- $W_3$: The Gradient Centralization baseline might be under-tuned (fixed at $\beta=1$ without rescaling), although canonical. Exploring stronger GC variants could narrow the observed gap and yield a fairer comparison.

- $W_4$: SGMS is most effective with nonnegative activations (ReLU/SiLU); effects are smaller or neutral with zero-centered activations such as \texttt{tanh} or under short schedules, suggesting some activation dependence. As GeLU is prominent in nowadays NLP, it could be interesting to study if SGMS still compares favorably to vanilla training regime.

**Questions:**

- $Q_1$: Could the authors formalize the heuristic $v_1 \approx e$ assumption by providing a perturbation bound or a mild structural condition (e.g., approximate column homogeneity of $X^\top X$) under which the replacement $v_1 v_1^\top \to ee^\top$ is theoretically justified? I know this might be very technical to get to but as the justification in the paper is for now only empirical, I am worried this might not translate to other application fields (NLP, speech, etc.)

- $Q_2$: Have the authors considered extending SGMS to non-vision domains such as NLP? In particular, would the method improve optimization or stability when (i) pretraining a small GPT-2–like transformer on OpenWebText or (ii) fine-tuning a RoBERTa-style model on GLUE?

- $Q_3$: How sensitive is SGMS to the scaling factor $\beta$ in large-scale or long-training regimes? Do the authors observe consistent optimal values (e.g., $\beta \approx 0.9$), or does it vary across architectures and optimizers? Could the $\beta$ be tuned **during** training to avoid instabilities and to help convergence? Although the following may introduce additional cost and go somewhat beyond the paper’s scope, but I would be interested to know whether $\beta$ could in principle be learned automatically?

- $Q_4$: For activations like GeLU or other zero-centered functions, do the authors expect theoretical or empirical limitations for SGMS? Would a modified mean-subtraction scheme (e.g., per-feature normalization) alleviate this?

---

> ### Author Response · Authors · 2025-11-22
> **Responses to comment from 4W9V**
>
> We appreciate the constructive review.
>
> # Answer to Q1.
>
> We provide an additional theoretical justification for when the dominant empirical
> feature direction becomes nearly uniform.
> The key structural property required for this approximation is the non-negativity
> (or approximate non-negativity) of the activations.
> When X has nonnegative entries, $X^\top X$ becomes an entry-wise nonnegative matrix and
> the Perron–Frobenius theorem guarantees that its dominant eigenvector lies in the
> positive orthant, ensuring strictly positive correlation with the uniform direction $e$.
> This argument does not depend on the specific application domain and therefore applies
> whenever the activation statistics satisfy this condition.
>
> In addition, using the Davis–Kahan sinΘ theorem, we show that the leading eigenvector
> of the empirical Gram matrix remains close to e under mild deviations in per-column
> activation statistics.
> This provides a perturbation bound that formalizes when the replacement
> $v_1 \simeq e$ is theoretically justified, except under perturbations far larger
> than those observed empirically.
>
> The full derivation and discussion of the required conditions are provided in the
> Appendix H.
>
> # Answer to Q2.
>
> We evaluated SGMS on non-vision tasks by fine-tuning RoBERTa-base and DistilBERT
> on the GLUE benchmark.
> As reported in the revised manuscript (Experiment section in the main text), across the eight standard GLUE tasks
> (excluding WNLI which trained model easily underperforms most-frequent class prediction), SGMS is not observed to be harmful, and in several tasks
> it yields slight improvements for moderate β values.
> These mild effects are reasonable with the fact that pretrained language models
> already exhibit well-behaved feature geometry, leaving little room for SGMS to
> produce larger gains.
> Importantly, this also indicates that SGMS integrates stably into pretrained NLP pipelines without introducing optimization issues.
>
> # Answer to Q3.
>
> We examined the sensitivity of SGMS to the scaling factor $\beta$ using CIFAR-100,
> where a full sweep is feasible.  As shown in Appendix C3, SGMS exhibits stable improvements over vanilla baseline
> across a broad range ($\beta \in \{0.1,\cdots,1.0\}$), with comparatively high values (0.7–0.9)
> yielding maximal improvements.
> Using the CIFAR-selected value $\beta = 0.9$ on ImageNet-1K, we consistently observed
> stable training and accuracy gains across multiple architectures and optimizers,
> and did not encounter degradation.
>
> Regarding the question of whether the optimal $\beta$ is consistent across models:
> in principle, the best $\beta$ can vary with architecture, activation statistics, and
> optimizer dynamics.  However, our CIFAR-100 sweep indicates that SGMS is robust
> over a broad interval rather than depending on a sharply tuned optimum, which is
> why a single value ($\beta=0.9$) transferred reliably to ImageNet.
>
> As for adaptive $\beta$: one could attempt to choose $\beta$ using criteria such as maximizing the entropy of
> the gradient singular-value spectrum, but this would require estimating singular
> values during training and would violate the lightweight design goal of SGMS.
> One could also statically schedule $\beta$ during training, analogously to a
> learning-rate schedule. This is an interesting direction for future work,
> provided that the lightweight nature of SGMS can be preserved.
>
> # Answer to Q4.
>
> In practice, SGMS does not rely on activations being strictly nonnegative; what it requires is
> that the representations exhibit sufficient asymmetry so that a dominant
> mean-like direction appears in $X$. Although GeLU covers both positive and negative ranges,
> distribution is strongly skewed toward positive values, and thus typically
> satisfies this condition. Empirically, in a pretrained RoBERTa encoder we found
> that the leading singular vector of the GeLU activations aligns substantially
> with the uniform direction ($v_1^\top e = 0.8346$, please see Appendix I), indicating that a dominant
> direction is indeed present.  Consistent with this, SGMS provides clear
> improvements in DeiT, which also uses GeLU throughout.
>
> Our GLUE fine-tuning results, by contrast, show only small effects. This is
> expected, as pretrained language models already exhibit weak feature anisotropy
> and these tasks tend to saturate quickly, leaving little room for additional
> gradient-geometry modifications to produce measurable gains.  These observations
> do not suggest a fundamental limitation of SGMS under GeLU, but rather reflect
> the well-behaved representations of modern pretrained NLP models.

---

> > ### Comment · Reviewer_4W9V · 2025-11-25
> > **Answer to rebuttal**
> >
> > Thank you to the authors for the detailed and thoughtful rebuttal, as well as for the newly added analyses and experiments.
> >
> > Regarding $Q_1$, the additional discussion provides a clearer justification for when the approximation $v_1 \approx e$ is expected to hold. While I do not think it fully removes the heuristic nature of the assumption, it addresses my concern to a reasonable extent and clarifies the structural conditions under which the approximation is valid.
> > I have also skimmed through Appendix H and although it relies on idealized assumptions (which is fine), the mathematical derivation looks sound and provides a clear formal regime in which the approximation is justified.
> >
> >
> > Concerning $Q_2$, I appreciate the extra GLUE experiments on RoBERTa-base and DistilBERT. These results confirm that SGMS integrates stably into NLP fine-tuning pipelines and does not degrade performance. However, the improvements remain small or neutral, which suggests that the method offers limited benefit for pretrained language models. This is consistent with the authors’ explanation but does not materially strengthen the claims of broader generality beyond vision tasks.
> > While SGMS might not be useful for fine-tuning, due to well-behaved feature geometry of pre-trained models, it would be interesting to see if SGMS gives good results in pretraining models.
> > Empirical results on small pretraining tasks (GPT-2 small on WikiText-103 for example) could further strengthen the paper and the method.
> >
> > For $Q_3$, the sensitivity analysis of $\beta$ is helpful. The robustness observed across a wide range of values, and the successful transfer of $\beta=0.9$ from CIFAR-100 to ImageNet, provide evidence that SGMS does not require fine-grained tuning, at least on vision tasks.
> >
> > Regarding $Q_4$, the clarification about GeLU (and the empirical evidence that its activation statistics still induce a dominant mean-like direction) is useful. The explanation of why SGMS produces clearer gains in DeiT than in GLUE is consistent with the observed activation geometry differences between vision and pretrained NLP models.
> >
> > ---
> >
> > Overall, the rebuttal improves the paper and addresses several of my questions, particularly in terms of theoretical framing and stability across domains. At the same time, the additional experiments confirm that SGMS remains most effective in the vision setting, with limited gains in NLP.
> >
> > Having also read the other reviews and the corresponding rebuttals, and taking everything into account, I keep my original score for now.
> >
> > I thank again the authors for their work and would be very interested in seeing some Transformer pre-training results on standard NLP tasks.

---

> > > ### Author Response · Authors · 2025-12-02
> > >
> > > Thank you for the thoughtful follow-up comments.
> > >
> > > We agree that SGMS brings limited benefit in the fine-tuning regime of pretrained NLP models, which already exhibit well-behaved feature geometry.
> > >
> > > Motivated by the suggestion to evaluate SGMS in a pretraining setting, we conducted additional experiments on a compact GPT-style model trained from scratch on a 1M-token OpenWebText subset.
> > > As shown in the main text (we moved the GLUE experiments to Appendix), SGMS provides small but consistent improvements over the vanilla baseline across all values of $\beta$, with $\beta=0.7$ performing best.
> > > These results complement our vision experiments and demonstrate that SGMS remains stable and beneficial in language-model pretraining, even though the gains are modest compared to the vision models.

---

### Official Review · Reviewer_51Nx · 2025-11-03

**Soundness:** 3
**Presentation:** 2
**Contribution:** 2
**Rating:** 4
**Confidence:** 4

**Summary:**

The authors propose to center weight gradients during the update of gradient-descent trained neural networks (NN).

They motivate the update as an approximation of the update obtained by a rank-one correction on the original euclidean update, claimed to approximate the leading eigendirection of X^TX, which is empirically checked as the correlation between a numerically computed leading eigenvector (if I undestood correclty).

The update is benchmarked on ResNets and Transformers, showcasing some accuracy improvement over non-centered variants for a minimal overhead.

**Strengths:**

- Clear and accessible presentation, making the concepts easy to understand.
- Low additional implementation and computational overhead, with a versatile approach that can be integrated into various optimization algorithms such as Adam or momentum-based gradients.

**Weaknesses:**

1. The explanations in sections 3.2 and 3.3 are somewhat unclear. If I understand correctly, the ideal goal would be to perform the operation in Equation 6, which is subsequently approximated using Equation 9. How is the inverse-square root computed in Equation 6? This step does not seem to directly lead to the formulation in Equation 9, and clarification on this point would be helpful.
2. The technique of gradient centering has been known since the early days of neural network training. I have included some references below as examples. However, it also appears that such methods are not commonly implemented in standard neural network libraries like PyTorch, which might explain their limited popularity compared to heuristics like Adam. What novel insights or contributions does your paper provide in relation to these papers?
3. Based on my understanding of neural network optimization, the effects described are likely related to the interaction between bias vector updates and weight vector updates, through second-order effects or information geometry if motivating using the Natural Gradient. However, the paper does not discuss bias vectors at all. I would expect the impact of the proposed method to diminish in networks with small or negligible biases, and addressing this aspect could strengthen the analysis.

- Le Cun, Y., Kanter, I., & Solla, S. A. (1991). Eigenvalues of covariance matrices: Application to neural-network learning. Physical review letters, 66(18), 2396.9
- Schraudolph, N. N. (2002). Centering neural network gradient factors. In Neural Networks: Tricks of the Trade (pp. 207-226). Berlin, Heidelberg: Springer Berlin Heidelberg.
- Ollivier, Y. (2015). Riemannian metrics for neural networks I: feedforward networks. Information and Inference: A Journal of the IMA, 4(2), 108-153.

**Questions:**

4. Your method is motivated as an optimization technique, yet the results shown primarily report improvements in validation accuracy, if I am correct? Could you clarify this distinction? Do you also get improvement in training loss ?
5. How is the vector v_1 computed in Figure 1, which serves as baseline ?
6. What is the reasoning for considering the concentration of the update in a low-dimensional subspace as potentially undesirable, apart from claimed empirical improvement in validation accuracy ?
7. Lines 245–251 are somewhat imprecise, using phrases like "... often ...", "... some degree of similarity ...", and "... tends to align ...".

Finally, a minor comment that I don't expect to be addressed :)
- I disagree with the fact that performing power-iterations on X^TX would be too costly in practice (l243), especially given the fact that the spectrum of X^TX is highly imbalanced (this would require only a few, if not a single interation), and also since you can initiate the power iteration using your vector e. So this might be a lead for future improvement as well.

---

> ### Author Response · Authors · 2025-11-22
> **Responses to comments from 51Nx**
>
> We appreciate the constructive review.
>
> # Reply to W1.
>
> We first clarify the key point: SGMS does *not* compute the inverse square root in Eq. (6), nor does it directly approximate Eq. (6).
> The whitening-like operator in Eq. (6) is introduced only as a conceptual reference.
>
> Instead of computing this heavy operation, SGMS targets only its most influential effect: suppressing the contribution of the dominant singular direction of $X$.
> Under ReLU-like activations, the leading right singular vector $v_1$ is well aligned with the uniform vector $e$.
> This allows us to replace the inverse-square-root operation with the lightweight operator
>
> $P = I - \beta e e^{\top}$ (Eq.~(9)),
>
> which weakens the dominant direction without any matrix inverse or inverse square root computation.
>
> We will revise Section 3.3 to explicitly state that SGMS does not involve Eq. (6) computationally, and that Eq. (6) is used only to motivate the effect that SGMS reproduces in a simplified form.
>
> # Reply to W2.
>
> Thank you for pointing out these references. We have added them to the
> Related Work section in the revised manuscript.
>
> We acknowledge that the two ingredients underlying SGMS have each been
> discussed in prior work:
> -  LeCun et al. (1991) noted that suppressing the dominant singular
> direction can accelerate optimization when second-order information is
> available, and
> -  gradient centering (GC) has been observed to improve training
> stability in various settings.
>
> However, these lines of work remained conceptually separate.
> Our contribution is to show that, in modern networks with ReLU-like
> activations, these two ideas are deeply connected: the column-wise mean
> of $\Delta W$ closely approximates the dominant right singular direction of $X$.
> Thus, GC can be interpreted as implicitly removing this dominant
> direction.
>
> This geometric interpretation does not appear in prior work, and it is
> precisely this perspective that enables SGMS.
> Without recognizing the singular-direction structure, one would not
> arrive at the idea of partially suppressing the mean direction—retaining
> useful components while still smoothing the spectral skew of $X$.
>
> Therefore, while SGMS builds upon concepts that have been studied
> independently, the link between them—and the resulting design of a
> principled, tunable, and computationally trivial operator—is novel.
>
> We will clarify this connection in the revised manuscript.
>
>
> # Reply to W3.
>
> In our formulation, the bias term does not interact with the low-rank structure that motivates SGMS. For clarity, consider a
> single output channel $c$. Let $X \in
> \mathbb{R}^{n \times k}$, $y_c \in \mathbb{R}^n$, $w_{c} \in \mathbb{R}^k$, and $b$ represents the bias (scalar).
>
> The forward propagation can be written as
>
> $y_c = X w_c + b$,
>
> and the gradients are
>
> $\frac {\partial L} {\partial w_c} = X^\top \Delta y_c$,
>
> $\frac {\partial L} {\partial b} = \mathbf{1}^\top \Delta y_c$,
>
> where $\Delta y_c = \partial L / \partial y_c$ and $\mathbf{1}$ is the all-ones
> vector.
>
> The weight gradient is thus entirely controlled by the input matrix $X$ and
> the vector $\Delta y_c$, leading to the low-rank structure we analyze via
> $X^\top \Delta Y$. In contrast, the bias gradient depends only on the summed residual
> $\mathbf{1}^\top \Delta y_c$ and is independent of $X$. Therefore, the
> low-dimensional concentration we study arises only in the weight update, not in the
> bias update, and the presence or magnitude of the bias does not affect the phenomenon
> that SGMS aims to address.
>
> # Answer to Q4.
>
> Yes, SGMS improves not only validation accuracy but also the optimization dynamics.
> We have added training-loss curves for ImageNet models in the revised manuscript.
> Across ResNet-18, ResNet-50, and DeiT-Small, SGMS consistently reduces the
> training loss more rapidly than the Vanilla baseline while maintaining stable
> convergence. This confirms that SGMS is effective as an optimization technique, not
> only in terms of validation accuracy.
>
>
> # Answer to Q5.
>
> If we understand the question correctly, the direction shown for comparison in
> Figure 1 corresponds to the dominant direction of $X^\top X$ that appears in the
> derivation in Sec. 3.2. For the purpose of offline analysis in that figure, we calculated
> the first right singular vector of $X$ via SVD.
> This numerical estimate is used only for visualization; SGMS does not compute $v_1$
> during training and relies solely on the normalized mean direction $e$.

---

> > ### Author Response · Authors · 2025-11-22
> >
> > # Answer to Q6.
> >
> > Our point is not that low-dimensional concentration is universally harmful, but that
> > in modern deep networks it can be unexpectedly strong (Figure 1), and that reducing
> > this imbalance with SGMS activates additional descent directions that would otherwise
> > be under-utilized. SGMS can be viewed as a lightweight mechanism that partially
> > mimics the geometric effect of preconditioned methods such as AdaBK, without maintaining any
> > second-order structure. Empirically, this leads to faster reduction in training loss
> > and improved validation accuracy, suggesting that the optimizer benefits from access
> > to a richer set of update directions rather than being confined to a narrow
> > low-dimensional subspace.
> >
> > A complementary perspective comes from covariance-based preconditioners such as
> > AdaBK, which show that flattening the singular-value spectrum can improve the
> > regret bounds of gradient-based optimization.
> > While SGMS does not attempt to fully equalize all singular values, suppressing
> > only the dominant component already moves the update toward a flatter spectrum.
> > This provides a principled rationale—consistent with the AdaBK analysis—for why
> > the lightweight correction introduced by SGMS can lead to more effective descent.
> >
> > # Answer to Q7.
> >
> > We have revised those lines to remove the imprecise expressions and to make the
> > reasoning mathematically precise.
> >
> > In the updated manuscript, we state explicitly that when activations are
> > nonnegative, the matrix $X^\top X$ is element-wise nonnegative and the
> > Perron–Frobenius theorem guarantees that its dominant eigenvector lies in the
> > positive orthant.
> > This ensures a strictly positive correlation with the uniform direction e,
> > providing a formal justification for using e as a surrogate for the leading
> > right singular vector.
> > We also added a mathematical discussion in the Appendix H analyzing the robustness
> > of this approximation under perturbations, in response to comments from other
> > reviewers.
> >
> >
> > # Answer to "I disagree with the fact that performing power-iterations ... improvement as well."
> >
> > We appreciate the suggestion.
> >
> > We agree that, given the spectral imbalance of $X^\top X$, a power iteration
> > initialized at $e$ would converge in only a few steps.
> > However, the practical limitation is not the iteration count but the memory
> > overhead incurred by applying $X^\top X$ during training.
> >
> > Even a multiplication by $X^\top X$ (whether formed explicitly or via the
> > composition $X^\top(Xv)$) introduces intermediate tensors whose size scales with
> > the channel dimension $C$ and the batch/spatial dimensions.
> > For modern convolutional and Transformer layers, these intermediate tensors are large
> > enough that the additional memory footprint becomes non-negligible in practice.
> >
> > In addition, from an implementation standpoint, frameworks such as PyTorch
> > discard intermediate activations like $X$ once their contribution to the
> > backpropagation has been used.
> > Performing any operation involving $X^\top X$ would therefore require keeping
> > $X$ alive in a safe memory region outside the backward pass, which
> > dramatically increases memory consumption.
> > Avoiding such activation retention is precisely why SGMS is designed to operate
> > only on quantities already available during the gradient update.
> >
> > While exploring more accurate yet memory-efficient approximations could indeed be
> > an interesting direction for future work, directly running power iterations on
> > $X^\top X$ is incompatible with the lightweight nature that makes SGMS practical.

---

> > > ### Comment · Reviewer_51Nx · 2025-11-25
> > >
> > > Thanks for your response. Regarding W2 and the d3aU review, I don't believe the paper demonstrates a different effect than that reported in the provided '90s papers. It is an approximation of the second-order interaction between the bias vector and the weight matrix, even if it is not explicitly computed. At the very least, an ablation study should be added, or better yet, theoretical arguments should be included.
> > >
> > > I appreciate your consideration of my other remarks. I believe these improvements will strengthen the paper. However, in its current state, I still believe it lacks a more thorough discussion of its connection with prior works, which is necessary for the paper to meet the publication standards for ICLR.

---

> > > > ### Author Response · Authors · 2025-12-02
> > > >
> > > > Thank you for the follow-up comments.
> > > >
> > > > We would like to clarify that your interpretation aligns with our original intention: SGMS is indeed designed as a lightweight approximation of the second-order interaction.
> > > >
> > > > Our initial submission may not have emphasized this connection strongly enough, which likely caused the misunderstanding. The aim of SGMS is not to introduce a new optimization effect, but rather to provide a new and extremely lightweight realization of this well-understood effect that is compatible with modern ReLU/GeLU networks.
> > > >
> > > > The classical approach in '90s achieves this effect via forward-pass modifications based on analysis of explicit second-order statistics, whereas SGMS recovers it through a simple mean subtraction in backward-pass.
> > > >
> > > > We will revise the text to make this positioning more explicit.

---

### Author Response · Authors · 2025-11-14
**Response to Reviewers**

We sincerely thank all reviewers for their thorough and constructive feedback.

We are currently preparing detailed responses to each comment, including additional analysis and experiments where feasible.
We will provide point-by-point replies shortly.

---

### Meta-Review · Area_Chair_rrnM · 2026-01-09

**Summary:**

The paper proposes Scaled Gradient Mean Subtraction (SGMS), a gradient regularization technique to mitigate the low-rank concentration of weight updates in deep networks. By subtracting a scaled mean from the gradients, the method approximates spectral whitening under the assumption of non-negative activations. While reviewers appreciated the method's simplicity, clarity, and low computational overhead, there was a consensus that the novelty is limited. Specifically, the method is mathematically very similar to Gradient Centralization (GC), with the primary innovation being a tunable scalar parameter $\beta$ for partial suppression. Reviewers also expressed concern over the marginal nature of the empirical gains and the method's limited effectiveness outside of computer vision tasks.

**Reviewer Concerns:**

*** Addressed
The authors addressed concerns regarding the heuristic justification of the method ($v_1 \approx e$). By providing additional theoretical justification using the Perron-Frobenius theorem and Davis-Kahan bounds in the appendix, they satisfied Reviewer 4W9V. The authors also addressed the lack of non-vision experiments (raised by 4W9V and d3aU) by providing GLUE fine-tuning and GPT-style pre-training results, confirming the method is stable though less effective in NLP. Missing comparisons to GC on ImageNet (requested by d3aU) were also added.

*** Outstanding
The primary concern regarding novelty remains largely unresolved. Reviewers 51Nx, G3oa, and d3aU viewed the method as a minor variation of Gradient Centralization or classical centering techniques, arguing that the shift from full $\beta=1$ to partial $\beta < 1$ suppression is a hyperparameter tune rather than a distinct methodological contribution. Furthermore, the empirical evidence did not convincingly demonstrate that this partial suppression yields significant practical benefits over the existing GC baseline, with improvements often being marginal $<1\%$. Reviewer 51Nx specifically noted that the connection to second-order interactions and bias vectors was still not sufficiently distinguished from prior literature.

**Reviewer Scores:**

Reviewer 4W9V: Maintained their score of 6. This reviewer explicitly stated in the discussion that while the rebuttal improved the paper's theoretical framing and robustness checks, the limited gains in NLP confirmed the method is primarily vision-focused, leading them to keep their original assessment.

Reviewer 51Nx: Maintained a score of 4. In their final comment, they acknowledged the clarifications but reiterated that the paper lacked a thorough discussion of its connection to prior works (specifically 1990s centering papers) and maintained that the method does not demonstrate a fundamentally different effect.

Reviewer G3oa: Maintained a score of 4. While the authors clarified technical questions about activation assumptions, the reviewer's core critique (that the method is a minor conceptual modification of GC with limited performance gains) was not fundamentally altered by the additional experiments.

Reviewer d3aU: Maintained a score of 2. This reviewer had strong reservations about the utility of the method compared to modern covariance-based preconditioners and viewed the similarity to GC as a critical flaw. While the authors added requested baselines, the fundamental criticism regarding the method's novelty and the necessity of the extra hyperparameter stands.

---

### Decision · Program_Chairs · 2026-01-26

Reject